# Rethinking 3D Shape Generation: Diffusion over Superquadrics

**Zhiyang Liu** [1 2]  **Wanze Li** [2]  **Yuwei Wu** [2]  **Chengran Yuan** [1 2]  **Jiawei Sun** [1 2]  **Rui Zheng** [3]  **Marcelo H Ang Jr** [1 2]

## Abstract

Diffusion models have advanced 3D shape generation, yet most methods still denoise in high-cardinality spaces (e.g., voxel/SDF grids, meshes, or point clouds), which is computationally and memory intensive and makes it difficult to scale in terms of both higher resolution and stronger controllability. We rethink the diffusion representation and propose to move diffusion from dense geometry to compact geometric primitives, representing each shape as a small set of **superquadrics**. Instead of operating on thousands to millions of geometric representation values, we leverage 7KB superquadric parameters (pose, size, and shape), drastically reducing diffusion-state dimensionality and per-step compute/memory. Our diffusion-over-superquadrics improves scalability by supporting broader capabilities (e.g., resolution-free point-cloud decoding, part-level editing, and constraint-based design) and achieving competitive surface-fidelity and distributional performance on standard benchmarks after point-cloud decoding, while enabling efficient generation within 0.6s per shape for most conditions.

## 1. Introduction

Diffusion models generate samples by learning the denoising process (Ho et al., 2020; Song et al., 2020), and recent Transformer-based designs have improved scalability to larger models and datasets (Peebles & Xie, 2023; Bao et al., 2023). This progress has enabled 3D diffusion for shape generation, from unconditional to image-to-3D settings (Wang et al., 2025). Yet a central bottleneck remains:

*the representation that diffusion is asked to model.*

Most 3D diffusion methods denoise high-cardinality representations. Point-based models denoise thousands of continuous values per shape (e.g., 2,048-point clouds are standard in ShapeNet benchmarks) (Zeng et al., 2022; Chang et al., 2015), while grid-based discretizations can be far larger: sparse voxel generators explicitly target millions of voxels (Ren et al., 2024a), and tetrahedral partitions scale cubically with resolution (e.g., $0.72 \cdot R^3$ elements, evaluated up to $R{=}192$) (Kalischek et al., 2024). Because denoising is repeated for tens to hundreds of steps, compute, memory, and runtime scale sharply with the dimensionality of the diffusion state. Moreover, when the state itself is high-cardinality, model capacity is often spent on dense geometric bookkeeping rather than object-level structure. In response, recent lines of work improve efficiency by changing the diffusion space, e.g., via hierarchical discretizations such as octrees (Xiong et al., 2025) or compact latent parameterizations such as tri-planes (Wu et al., 2024). However, these approaches expose a familiar trade-off: discretization-based methods remain resolution-dependent, while latent diffusion often sacrifices explicit geometric interpretability and direct editability for compactness. This leads to a natural question:

> Can we scale diffusion-based 3D generation while retaining an explicit and compact representation?

We answer this question by rethinking the diffusion representation. Rather than denoising dense geometric fields, we perform **diffusion over superquadrics** (**DoSs**), representing each shape as a set of geometric primitives with continuous parameters for shape ($\mathbb{R}^2$), size ($\mathbb{R}^3$), and pose ($SE(3)$). This primitive parameter representation is explicit. Each token corresponds to an interpretable geometric component. It is also compact because DoSs denoises a fixed size token representation per shape, instead of denoising geometry at a chosen voxel resolution or point count. Recent work has revisited superquadrics as practical building blocks for part structured 3D representations and structured reasoning at scale (Fedele et al., 2025; Zuo et al., 2025). This motivates using superquadrics as generative tokens in DoSs. 3D generation then becomes denoising a short token sequence, encouraging the model to capture part structure and inter part relationships directly.

---

[1]Advanced Robotics Centre, National University of Singapore [2]College of Design and Engineering, National University of Singapore [3]Engineering Systems and Design, Singapore University of Technology and Design. Correspondence to: Zhiyang Liu <iszhiyang@gmail.com>.

*Proceedings of the $43^{rd}$ International Conference on Machine Learning*, Seoul, South Korea. PMLR 306, 2026. Copyright 2026 by the author(s).

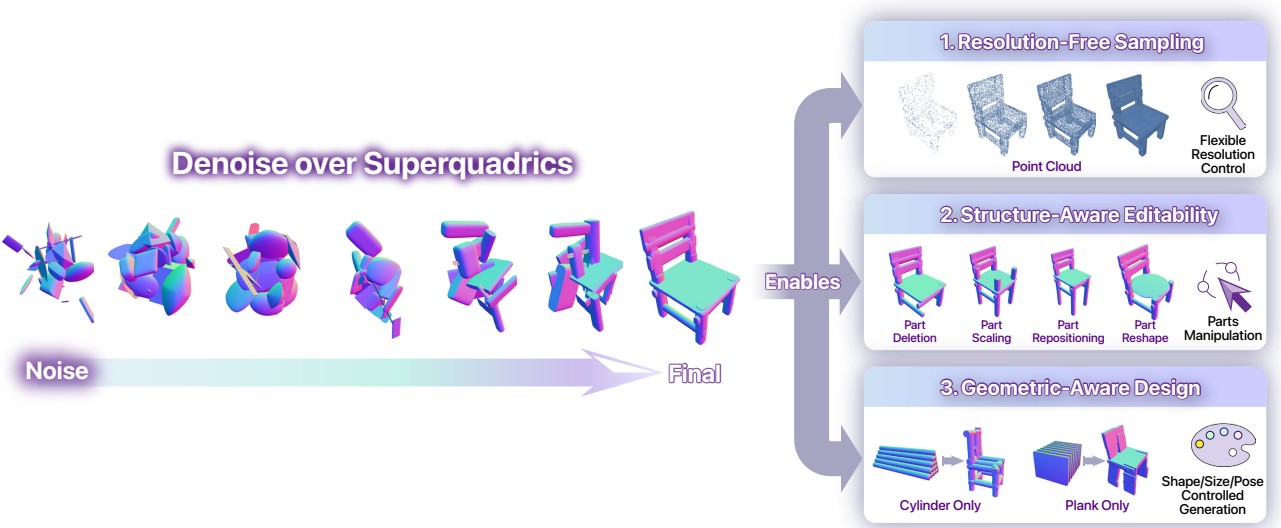

*Figure 1.* Denoise over superquadrics. Starting from noise, our model denoises a compact set of superquadric primitives to a final structured shape, which can be decoded into point clouds at arbitrary resolution. The explicit primitive parameters further enable structure-aware editability (part deletion, scaling, repositioning, and reshaping) and geometric-aware design via constrained denoising process (e.g., cylinder-only or plank-only generation).

Based on this perspective, we introduce a diffusion model that generates superquadric parameters directly, but a naive implementation is not straightforward for three reasons. First, shapes require different numbers of primitives depending on geometric complexity, so we represent each shape by up to $K_{\max}$ primitives and include an additional existence score per primitive to handle variable primitive counts. Second, diffusion perturbs variables with Gaussian noise in an ambient Euclidean space (Ho et al., 2020; Song et al., 2020), but pose includes rotation on $SO(3)$, which is a curved manifold and can be discontinuous under common parameterizations, so we follow Zhou et al. (2019) and use a continuous 6D rotation representation mapped to a valid rotation matrix via orthonormalization. Third, our diffusion model (Ho et al., 2020) uses a 1D convolutional U-Net backbone that operates on a fixed length ordered token tensor, so we impose a deterministic canonical ordering of primitives, by volume or position, to convert an unordered primitive set into a consistent token sequence for learning. Fig. 1 overviews DoSs. Starting from Gaussian noise, our model denoises a compact set of superquadric primitives into a structured shape, which is then decoded into point clouds at arbitrary sampling resolution. The explicit primitive parameters also support part level edits and constrained denoising for design. Our contributions are threefold.

- **Rethinking shape representation for diffusion.** We rethink 3D diffusion by shifting the denoising target from high cardinality voxel, mesh, or point representations to a compact and explicit set of superquadric primitives, and propose diffusion over superquadrics (DoSs) for 3D shape generation.

- **Making superquadrics diffusable.** We develop a practical diffusion formulation over superquadric primitives by addressing three implementation obstacles: variable primitive counts are handled with an existence score, rotations are diffused with a continuous 6D representation (Zhou et al., 2019), and unordered primitives are converted into a consistent token sequence using deterministic canonical ordering.

- **Efficiency, quality, and scalable capabilities.** We show that DoSs achieves competitive fidelity and distributional performance on ShapeNet (Chang et al., 2015), while improving efficiency through a much smaller diffusion state. We further demonstrate capabilities enabled by DoSs, including resolution-free decoding, part level edits, and constraint based design via constrained denoising.

## 2. Related Work

**Diffusion for 3D shape generation.** Recent 3D diffusion methods differ largely by the representation they denoise. A line of work performs diffusion over explicit 3D samples or discretizations such as point sets and voxelized tokens. Point E generates point clouds via cascaded diffusion with image conditioning (Nichol et al., 2022), and DiT 3D shows that diffusion transformers can denoise voxelized point cloud tokens effectively (Mo et al., 2023). These explicit approaches provide straightforward geometry extraction and evaluation, but their generation cost often grows with sampling density or grid resolution, and enforcing part level edits or hard structural constraints typically requires additional design of conditioning mechanisms.

Another line of work improves scalability by moving diffusion to compact latent or implicit spaces and decoding to surfaces. Sin3DM and Direct3D use triplane latent representations and perform diffusion in that latent space (Wu et al., 2023b; 2024). While such methods can decouple diffusion from dense 3D grids, the intermediate variables are often not directly interpretable, which makes explicit geometric control and constraint enforcement less direct.

**Superquadrics and primitive representations.** Superquadrics are a classical family of parametric primitives that compactly represent a range of quadric like shapes (Barr, 1981). Recent work revisits superquadrics as interpretable parts for reconstruction and decomposition. SuperDec decomposes objects and scenes into superquadric primitives from point clouds (Fedele et al., 2025), ISCO constructs superquadric based object models from multiple views (Alaniz et al., 2023), and Differentiable Blocks World represents scenes as textured superquadric meshes optimized through differentiable rendering (Monnier et al., 2023). These methods focus on reconstruction, fitting, or decomposition rather than learning a category level generative prior. In contrast, we perform diffusion directly over a set of superquadric primitives, yielding a compact and explicit denoising space that supports resolution free decoding and direct parameter level edits and constraints.

## 3. Methodology

We represent each object as a set of superquadric primitives and learn a denoising diffusion model directly in this compact parameter space. The methodology has four parts: (i) revisiting superquadrics, (ii) revisiting Diffusion Probabilistic Models (DDPM), (iii) what makes superquadrics diffusable in practice, and (iv) how to scale diffusion over superquadrics to broader capabilities.

### 3.1. Revisiting Superquadrics

**Superquadrics as geometric primitives.** Superquadrics are a family of compact geometric primitives that interpolate smoothly between round and boxy shapes, making them a flexible continuous representation for 3D geometry. Fig. 2 illustrates this effect by varying the two shape exponents $(\epsilon_1, \epsilon_2)$ while keeping pose and scale fixed. Let $\mathbf{x} = (x, y, z)^\top \in \mathbb{R}^3$ denote a point in the local coordinate frame of a superquadric. A superquadric is parameterized by axis lengths $\mathbf{a} = (a_1, a_2, a_3)^\top$ and shape exponents $\boldsymbol{\epsilon} = (\epsilon_1, \epsilon_2)^\top$, where $a_i > 0$ and $\epsilon_j > 0$. Its inside-outside function is shown in Eq. 1. The surface is given by $\{\mathbf{x} \mid F(\mathbf{x}; \mathbf{a}, \boldsymbol{\epsilon}) = 1\}$. Points with $F(\mathbf{x}; \mathbf{a}, \boldsymbol{\epsilon}) < 1$ lie inside the shape, and those with $F(\mathbf{x}; \mathbf{a}, \boldsymbol{\epsilon}) > 1$ lie outside.

$$F(\mathbf{x}; \mathbf{a}, \boldsymbol{\epsilon}) = \left( \left| \frac{x}{a_1} \right|^{\frac{2}{\epsilon_2}} + \left| \frac{y}{a_2} \right|^{\frac{2}{\epsilon_2}} \right)^{\frac{\epsilon_2}{\epsilon_1}} + \left| \frac{z}{a_3} \right|^{\frac{2}{\epsilon_1}}. \quad (1)$$

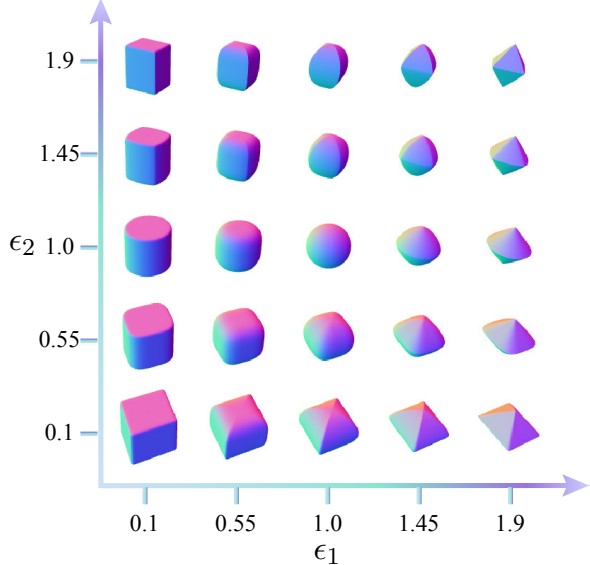

*Figure 2.* A grid of superquadrics generated by varying the two shape exponents $(\epsilon_1, \epsilon_2) \in \{0.1, 0.55, 1.0, 1.45, 1.9\}^2$ (axes), while keeping pose and scale fixed.

**Pose in $SE(3)$.** Each primitive has a rigid pose $(\mathbf{R}, \mathbf{t})$ with $\mathbf{R} \in \mathrm{SO}(3)$ and $\mathbf{t} \in \mathbb{R}^3$. For a point $\mathbf{p} \in \mathbb{R}^3$ in the world frame, local coordinates are $\mathbf{x} = \mathbf{R}^\top(\mathbf{p} - \mathbf{t})$, and the posed implicit surface is $F(\mathbf{R}^\top(\mathbf{p} - \mathbf{t}); \mathbf{a}, \boldsymbol{\epsilon}) = 1$.

**Superquadrics volume (for canonical ranking).** We use a closed-form volume for superquadrics (Jaklic et al., 2000):

$$\mathrm{Vol}(\mathbf{a}, \boldsymbol{\epsilon}) = 2a_1 a_2 a_3 \, \epsilon_1 \epsilon_2 \, \mathrm{B}(\frac{\epsilon_1}{2}, \epsilon_1 + 1) \mathrm{B}(\frac{\epsilon_2}{2}, \frac{\epsilon_2 + 2}{2}). \quad (2)$$

Here $\mathrm{B}(p, q) = \Gamma(p)\Gamma(q)/\Gamma(p + q)$, where the Gamma function is defined for $t > 0$ as $\Gamma(t) = \int_0^\infty u^{t-1} e^{-u} \, du$.

### 3.2. Revisiting DDPM

In this work, we adopt DDPM (Sohl-Dickstein et al., 2015; Ho et al., 2020) to generate new data $\mathbf{x}_0$ that obey the distribution of the training set. Concretely, DDPM recovers the output $\mathbf{x}_0$ from a Guassian noise $\mathbf{x}_T$ by iteratively denoising with a noise prediction network $\boldsymbol{\xi}_\theta$

$$\mathbf{x}_{t-1} = \gamma_t(\mathbf{x}_t - \beta_t \boldsymbol{\xi}_\theta(\mathbf{x}_t, t)) + z_t, \quad (3)$$

Here $t = T, T - 1, ..., 1$ is the index for the iteration. $z_t$ is a Gaussian noise with zero mean and $\gamma_t, \beta_t$ are parameters determined by timestep.

The DDPM is trained in the forward (noising) process, where the Gaussian noise determined by $t$ is added to the training data iteratively $q(\mathbf{x}_t \mid \mathbf{x}_{t-1}) = \mathcal{N}(\sqrt{\alpha_t}\mathbf{x}_{t-1}, (1 - \alpha_t)\mathbf{I})$, which implies

$$\mathbf{x}_t = \sqrt{\bar{\alpha}_t}\mathbf{x}_0 + \sqrt{1 - \bar{\alpha}_t}\boldsymbol{\xi}. \quad (4)$$

where $\boldsymbol{\xi} \sim \mathcal{N}(\mathbf{0}, \mathbf{I})$, $\bar{\alpha}_t = \prod_{s=1}^{t} \alpha_s$, and $\alpha_t$ is linearly determined by $t$ in this work. The noise prediction network $\boldsymbol{\xi}_\theta$ is trained to learn the noise term in Eq.4 and the simplified training objective is

$$\mathcal{L}_{\text{DDPM}} = \mathbb{E}_{t, \mathbf{x}_0, \boldsymbol{\xi}} \left[ \|\boldsymbol{\xi} - \boldsymbol{\xi}_\theta(\mathbf{x}_t, t)\|_2^2 \right]. \qquad (5)$$

As proven in (Ho et al., 2020), minimizing Eq.5 is equivalent to minimizing the KL-divergence between the actual distribution of the training set and the distribution of data generated using Eq.3.

### 3.3. What Makes Superquadrics Diffusable?

This section presents three design choices that make diffusion over superquadrics well defined and practical: (i) existence scores to handle variable primitive counts, (ii) a continuous 6D rotation parameterization for pose diffusion, and (iii) deterministic canonical ordering to obtain a consistent token sequence.

#### 3.3.1. TOKENIZATION VIA EXISTENCE SCORES

**Variable primitive counts.** Different objects are decomposed into different numbers of superquadric primitives. Let object $i$ contain $K_i$ primitives, where $K_i$ varies across the dataset. Since DDPM requires a fixed-dimensional state, we fix a global maximum $K_{\max}$ and represent every object with $K_{\max}$ slots. To retain the original cardinality, each slot carries an existence score.

**Token with existence.** For slot $k$, we store translation $\mathbf{t}_k \in \mathbb{R}^3$, a 6D rotation parameter $\mathbf{r}_k \in \mathbb{R}^6$ (decoded in Sec. 3.3.2), axis length $\mathbf{a}_k \in \mathbb{R}^3$, unconstrained shape variables $\boldsymbol{\epsilon}_k \in \mathbb{R}^2$, and an existence score $e_k \in \mathbb{R}$:

$$\tilde{\mathbf{z}}_k = \left[ \mathbf{t}_k^\top, \mathbf{r}_k^\top, \mathbf{a}_k^\top, \boldsymbol{\epsilon}_k^\top, e_k \right]^\top \in \mathbb{R}^{15}. \qquad (6)$$

**Fixed-size state.** We pad each sample to $K_{\max}$ rows using all-zero rows. For each slot, we set $e_k = 1$ if it corresponds to a valid primitive and $e_k = 0$ otherwise, producing a fixed-size matrix $\tilde{\mathbf{Z}}_0 \in \mathbb{R}^{K_{\max} \times 15}$. This yields a Euclidean diffusion state while preserving object-specific cardinality through the existence channel. In Sec. 4.3, we ablate this design against a copy-paste padding baseline that fills missing slots by duplicating existing primitives to reach $K_{\max}$.

#### 3.3.2. SO(3) DIFFUSION VIA 6D ROTATIONS

**Continuity and ambiguity.** DDPM assumes the state evolves smoothly in Euclidean coordinates, but common rotation parameterizations are not globally well-behaved. Euler angles satisfy periodic identifications, e.g., $\mathbf{R}(\boldsymbol{\theta}) = \mathbf{R}(\boldsymbol{\theta} + 2\pi \mathbf{e}_j)$, and unit quaternions exhibit the antipodal symmetry $q \sim -q$. Such non-unique / discontinuous coordinates can cause the forward noising process to mix distinct rotations and make denoising unstable.

**6D rotations.** Primitive orientations are stored as ZYX Euler angles $\boldsymbol{\phi} \in \mathbb{R}^3$. Since diffusion perturbs variables with Gaussian noise in an ambient Euclidean space, we avoid diffusing Euler angles directly and instead diffuse the continuous 6D rotation representation of Zhou et al. (2019). We first convert Euler angles to a rotation matrix $\mathbf{R} = \mathcal{R}_{\text{ZYX}}(\boldsymbol{\phi}) \in \text{SO}(3)$, then map $\mathbf{R}$ to a 6D vector by stacking its first two columns: $\mathbf{a}_1 = \mathbf{R}_{:,1}$, $\mathbf{a}_2 = \mathbf{R}_{:,2}$, and $\mathbf{r} = [\mathbf{a}_1; \mathbf{a}_2] \in \mathbb{R}^6$. During training and sampling, diffusion is applied to $\mathbf{r}$ in $\mathbb{R}^6$. When forming poses or evaluating geometry, we decode $\mathbf{r}$ back to a valid rotation matrix by Gram Schmidt orthonormalization. Specifically, we normalize $\mathbf{a}_1$ to obtain $\mathbf{b}_1 = \text{norm}(\mathbf{a}_1)$, remove the component of $\mathbf{a}_2$ along $\mathbf{b}_1$ and normalize to obtain $\mathbf{b}_2 = \text{norm}(\mathbf{a}_2 - (\mathbf{b}_1^\top \mathbf{a}_2)\mathbf{b}_1)$, then set $\mathbf{b}_3 = \mathbf{b}_1 \times \mathbf{b}_2$. The decoded rotation matrix is

$$\mathbf{R}(\mathbf{r}) = [\mathbf{b}_1 \ \mathbf{b}_2 \ \mathbf{b}_3] \in \text{SO}(3). \qquad (7)$$

This construction guarantees $\mathbf{R}(\mathbf{r})$ is orthonormal with determinant 1, while keeping the diffusion variable $\mathbf{r}$ in a continuous Euclidean space. We ablate Euler angle diffusion versus the 6D representation in Sec. 4.3.

#### 3.3.3. SUPPRESSING PERMUTATION

**Permutation ambiguity.** Even after fixing the tensor size, the superquadric representation of an object is fundamentally a set: permuting primitive indices does not change the represented shape. However, our diffusion model (Ho et al., 2020) operates on an ordered token tensor $\tilde{\mathbf{Z}}_0$. If the same object appears with arbitrary permutations, the data distribution in token space contains multiple equally valid orderings, which can hinder learning: $\{\tilde{\mathbf{z}}_1, \ldots, \tilde{\mathbf{z}}_K\} \equiv \{\tilde{\mathbf{z}}_{\pi(1)}, \ldots, \tilde{\mathbf{z}}_{\pi(K)}\}$, $\forall \pi \in \mathcal{S}_K$, where $\mathcal{S}_K$ is the set of all permutations of $K$ elements. This permutation ambiguity injects ordering noise into denoising and can lead to inconsistent token identities across samples.

**Deterministic canonicalization.** We impose a deterministic primitive ordering before padding so each shape maps to a consistent token tensor. We consider two strategies and ablate them in Sec. 4.3. (i) Volume based: compute $V_k = \text{Vol}(\mathbf{a}_k, \boldsymbol{\epsilon}_k)$ using Eq. 2 and sort tokens by decreasing $V_k$. (ii) Position based: rank primitives by height using the $Y$ coordinate of their centers, $s_k = \mathbf{e}_y^\top \mathbf{t}_k = t_{k,2}$ with $\mathbf{e}_y = (0, 1, 0)^\top$, and sort by increasing $s_k$. Both reduce permutation variance and provide consistent supervision.

### 3.4. How to Scale Diffusion-over-Superquadrics?

Our superquadrics diffusion model outputs a compact primitive set in token form (Eq. 6), which enables several *diffusion-based generation tasks* beyond unconditional sampling. We focus on three capabilities that "scale" the generator in practice: (i) resolution-free surface sampling to point

clouds, (ii) structure-aware editability, and (iii) geometric-aware design via constrained denoising. Examples about these capabilities are displayed in Fig.1.

### 3.4.1. RESOLUTION-FREE SAMPLING

Given a generated tensor $\tilde{\mathbf{Z}}_0 \in \mathbb{R}^{K_{\max} \times 15}$, we first select valid primitives by thresholding the existence score:

$$\mathcal{K} = \{\, k \in \{1, \ldots, K_{\max}\} : e_k > \tau_e \,\}. \tag{8}$$

For each $k \in \mathcal{K}$, we decode the rotation matrix $\mathbf{R}_k = \mathbf{R}(\mathbf{r}_k)$ using Eq. (7).

**Per-primitive surface sampling.** We sample each primitive surface using the standard superquadric parameterization with the signed power operator $\mathrm{spow}(x, \epsilon) = \mathrm{sign}(x)\,|x|^\epsilon$. For angles $(\eta, \omega) \in [-\frac{\pi}{2}, \frac{\pi}{2}] \times [-\pi, \pi]$, the local surface point and its world coordinate are

$$\mathbf{x}_k(\eta, \omega) = \begin{bmatrix} a_{k1}\ \mathrm{spow}(\cos\eta, \epsilon_{k1})\ \mathrm{spow}(\cos\omega, \epsilon_{k2}) \\ a_{k2}\ \mathrm{spow}(\cos\eta, \epsilon_{k1})\ \mathrm{spow}(\sin\omega, \epsilon_{k2}) \\ a_{k3}\ \mathrm{spow}(\sin\eta, \epsilon_{k1}) \end{bmatrix},$$

$$\mathbf{p}_k(\eta, \omega) = \mathbf{R}_k\, \mathbf{x}_k(\eta, \omega) + \mathbf{t}_k.$$

Following the equal distance and arc length sampling procedure of Liu et al. (2022, Suppl., Sec. 4), we generate a candidate set $\mathcal{P}_k = \{\mathbf{p}_k(\eta_n, \omega_n)\}_{n=1}^{M_k}$ for each primitive and merge $\mathcal{P} = \bigcup_{k \in \mathcal{K}} \mathcal{P}_k$.

**Filtering points inside overlaps.** Because primitives can overlap, $\mathcal{P}$ may contain points that lie inside other primitives and should not appear on the surface of the composed shape. For a candidate point $\mathbf{p} \in \mathcal{P}_k$, we keep it only if it is outside every other primitive: $F_j\big(\mathbf{R}_j^\top(\mathbf{p} - \mathbf{t}_j); \mathbf{a}_j, \boldsymbol{\epsilon}_j\big) \geq 1$, $\quad \forall j \in \mathcal{K} \backslash \{k\}$, where $F_j(\cdot)$ is the inside outside function in Eq. (1). The remaining set is denoted by $\mathcal{P}^{\mathrm{surf}}$.

**Uniform density via FPS.** Arc length sampling is only near uniform on a single superquadric. In our setting, non uniformity arises from two sources: (i) the arc length sampler itself is approximate, and (ii) merging surface samples from multiple primitives, especially near overlaps and intersections, creates locally dense regions. To produce an even density point cloud at any requested size $N$, we run farthest point sampling (FPS) on the filtered surface set $\mathcal{P}^{\mathrm{surf}}$. Starting from any $\mathbf{p}_1 \in \mathcal{P}^{\mathrm{surf}}$, FPS iteratively adds $\mathbf{p}_n = \arg\max_{\mathbf{p} \in \mathcal{P}^{\mathrm{surf}} \backslash \mathcal{S}_{n-1}} \min_{\mathbf{q} \in \mathcal{S}_{n-1}} \|\mathbf{p} - \mathbf{q}\|_2$ and updates $\mathcal{S}_n = \mathcal{S}_{n-1} \cup \{\mathbf{p}_n\}$ until $|\mathcal{S}_N| = N$. This enables resolution free decoding, since the same generated superquadrics can be decoded into point clouds of arbitrary size without rerunning diffusion.

### 3.4.2. STRUCTURE-AWARE EDITABILITY

Superquadrics expose explicit part-level parameters (pose, size, shape) and an existence channel, so edits can be applied

directly in token space by modifying $\tilde{\mathbf{Z}}_0$. Typical operations include: (i) Part deletion: set $e_k \leftarrow \tau_e - \delta$ with $\delta > 0$ so slot $k$ is removed by Eq. 8; (ii) Part scaling: set $\mathbf{a}_k \leftarrow \boldsymbol{\lambda} \odot \mathbf{a}_k$ with $\boldsymbol{\lambda} \in \mathbb{R}_+^3$; (iii) Part reposing: set $\mathbf{t}_k \leftarrow \mathbf{t}_k + \Delta\mathbf{t}$ and overwrite $\mathbf{r}_k$ with the 6D encoding of a desired rotation $\mathbf{R}_k'$ (Sec. 3.3.2); (iv) Part reshaping: edit $\boldsymbol{\epsilon}_k$ to continuously control roundness and sharpness.

### 3.4.3. GEOMETRIC-AWARE DESIGN

Beyond post-hoc edits, superquadric diffusion supports design-time control by constraining specific parameters throughout denoising. Let $\Omega$ denote a set of geometric constraints expressed on token entries (or decoded parameters), and let $\Pi_\Omega$ be a projection operator that overwrites constrained channels with user-specified values. During ancestral sampling, we apply $\tilde{\mathbf{Z}}_{t-1} \leftarrow \Pi_\Omega(\hat{\tilde{\mathbf{Z}}}_{t-1})$ for $t = T, T-1, \ldots, 1$, which is a simple instance of constraint-incorporating diffusion sampling (Chung et al., 2022).

We take cylinder-only parts for furniture-like design as an example. Suppose we want a subset of primitives $\mathcal{K}_{\mathrm{cyl}} \subseteq \mathcal{K}$ to behave like "fixed-radius cylinders" (i.e., circular cross-sections with prescribed roundness). We can enforce $a_{k,1} = a_{k,2} = r_0$, $\boldsymbol{\epsilon}_k = \boldsymbol{\epsilon}^\star$, $\forall k \in \mathcal{K}_{\mathrm{cyl}}$, by clamping the corresponding entries. More generally, $\Omega$ can specify part templates (e.g., "all legs share the same radius"), symmetry constraints, or discrete part inclusion/exclusion through the existence channel. Crucially, these controls operate on explicit, low-dimensional primitive parameters, making the design constraints interpretable and easy to enforce during diffusion.

## 4. Experiments

### 4.1. Experimental Setup

**Dataset.** We follow the standard ShapeNet(Chang et al., 2015) used in prior work and report results on the CHAIR, AIRPLANE, and CAR categories. Given a ShapeNet mesh, we extract a set of superquadric primitives for training and testing using Marching-Primitives (Liu et al., 2023a), which fits superquadrics from a signed-distance representation.

**Evaluation metrics.** We follow prior ShapeNet generation works (Liu et al., 2023b; Mo et al., 2023) and report distributional metrics 1-NNA and Coverage (COV) under Chamfer Distance (CD). Since our DoSs outputs a set of superquadric tokens, we first decode each generated shape into a point cloud and evaluate in point space: we sample 2,048 surface points per shape (matching the standard evaluation resolution) and compute CD between point clouds. **1-NNA** is the 1-nearest-neighbor classification accuracy for distinguishing generated samples from real samples under CD; values closer to 50% are better, with 50% indicating indistinguishability. **COV** measures generation diversity by

*Table 1.* **Quantitative comparison on ShapeNet.** We report 1-NNA@CD ($\downarrow$, ideal $\approx 50$) and COV@CD ($\uparrow$) for Chair, Airplane, and Car. For each method, we list the diffusion state and its size (KB, float32), and inference time. Baseline 1-NNA/COV values are from DiT-3D (Mo et al., 2023) and TIGER (Ren et al., 2024b). Timing sources (RTX 3090, batch size 1): DPM/PVD from (Wu et al., 2023a), TIGER from (Ren et al., 2024b), LION from (Zeng et al., 2022), and DiT-3D is measured by us using the official checkpoint. Additional experimental results from multi-class training are provided in Appendix A. **Bold** and underline denote best and second best.

| Method | Diffusion state | Size (KB) | Inference (s/shape) | Chair 1-NNA | COV | Airplane 1-NNA | COV | Car 1-NNA | COV |
|---|---|---|---|---|---|---|---|---|---|
| DPM (Luo & Hu, 2021) | points (2048×3) | 24 | 22.8 | 60.05 | 44.86 | 76.42 | 48.64 | 68.89 | 44.03 |
| PVD (Zhou et al., 2021) | points (2048×3) | 24 | 29.9 | 57.09 | 36.68 | 73.82 | 48.88 | 54.55 | 41.19 |
| TIGER (Ren et al., 2024b) | points (2048×3) | 24 | 9.73 | 54.61 | – | 71.85 | – | 54.31 | – |
| LION (Zeng et al., 2022) | latent (128+8192) | 33 | 27.12 | 53.70 | 48.94 | 67.41 | 47.16 | 53.41 | **50.00** |
| MeshDiffusion (Liu et al., 2023b) | hybrid (64³×4) | 4096 | – | 53.69 | 46.00 | 66.44 | 47.34 | 81.43 | 34.07 |
| DiT-3D (Mo et al., 2023) | voxels (32³×3) | 384 | 12* | **49.11** | **52.45** | 62.35 | **53.16** | **48.24** | 50.00 |
| Ours | superquadrics (128×15) | **7** | **0.6** | 53.80 | 51.31 | **61.92** | 46.77 | 57.50 | 49.26 |

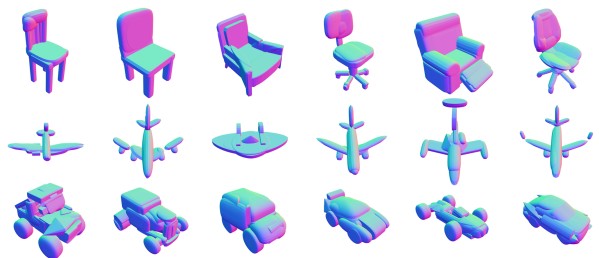

*Figure 3.* Qualitative results from DoSs. Unconditional generations on ShapeNet categories (top to bottom: chair, airplane, car). More visualization for denoising and generation is shown in Appendix E.

the fraction of real shapes that are covered by the generated set under nearest neighbor matching, where a real shape is counted as covered if it is selected as the nearest neighbor of at least one generated shape; higher is better.

**Implementation details.** We train category-specific models for 4,000 epochs using Adam with learning rate $1 \times 10^{-4}$ and batch size 64. Owing to the compact superquadric token diffusion space, training completes in about **2 hours per category** on a single RTX 4090 with **2 GB VRAM usage**. At inference, we run 100 denoising steps and obtain a per-shape denoising time of **0.6 s** with batch size 1. For each object, the $K_{\max}$ is set as 128 and $\tau_e = 0.5$.

### 4.2. Comparison with Existing Diffusion Models

Tab. 1 compares DoSs with representative diffusion baselines on ShapeNet CHAIR, AIRPLANE, and CAR, using standard distributional metrics 1-NNA@CD and COV@CD computed on 2,048-point clouds. Fig. 3 shows unconditional samples generated by DoSs across the three categories.

**Comparison scope.** We emphasize that this comparison focuses on unconditional ShapeNet generation under a matched point-cloud evaluation protocol. Several recent 3D generation systems target different settings, such as text-to-3D or image-to-3D, and differ in conditioning signals, training scale, output modality, and evaluation protocol. Direct

numerical comparison with these systems would therefore conflate the effect of representation choice with conditioning strength and data scale. Accordingly, our goal is not to claim state-of-the-art visual quality across all 3D generation settings, but to isolate the efficiency–controllability–quality trade-off obtained by moving diffusion into an explicit and compact geometric primitive space.

**Quality and diversity.** Methods that diffuse dense 3D states still provide the strongest distribution matching. DiT-3D achieves near-ideal 1-NNA on CHAIR and CAR, and the best COV on CHAIR and AIRPLANE, reflecting the fidelity advantage of voxel-space denoising. In contrast, point-space diffusion methods (DPM, PVD, TIGER) denoise $2048 \times 3$ unconstrained coordinates and often suffer from larger distribution gaps, particularly on AIRPLANE. LION narrows this gap by moving diffusion to a learned latent space, giving a more balanced 1-NNA/COV trade-off. DoSs occupies a different point in this spectrum: it sacrifices some dense-geometry fidelity for compactness and efficiency, while remaining competitive in distributional quality. It achieves strong 1-NNA, including the best result on AIRPLANE, and high COV on CHAIR, though its COV on AIRPLANE and CAR still trails the strongest dense-state baseline.

**Efficiency and diffusion state.** Runtime is largely governed by the size of the denoised state. Point diffusion denoises $\sim 24$ KB states ($2048 \times 3$ float32), DiT-3D voxel diffusion uses $\sim 384$ KB states ($32^3 \times 3$), and mesh-hybrid diffusion can be even larger (e.g., $\sim 4096$ KB for $64^3 \times 4$). In contrast, DoSs denoises only $128 \times 15$ superquadric tokens, or $\sim 7$ KB. This compact state directly improves sampling efficiency, yielding 0.6 s per shape at batch size 1 in our setup (Tab. 1). These results support our central message: moving diffusion from dense geometric fields to compact superquadric tokens offers an effective efficiency–quality trade-off for unconditional 3D generation at standard evaluation resolution. Qualitative samples and denoising visualizations are shown in Fig. 3 and Appendix E.

*Table 2.* Ablations on CHAIR under CD-based metrics. We ablate existence token, 6D orientation representation, and canonical sorting (volume-rank vs. $y$-rank).

| Exist. token | 6D orient. repr. | Vol.-rank | $y$-rank | 1-NNA ↓ | COV ↑ |
|:---:|:---:|:---:|:---:|:---:|:---:|
| × | × | × | × | 93.32 | 19.91 |
| ✓ | × | × | × | 58.36 | 48.41 |
| ✓ | ✓ | × | × | 57.57 | 50.37 |
| ✓ | ✓ | × | ✓ | 55.28 | 50.93 |
| ✓ | ✓ | ✓ | × | **53.80** | **51.31** |

**Multi-class training.** We further evaluate a single multi-class DoSs model trained jointly on CHAIR, AIRPLANE, and CAR using category conditioning; the detailed results are reported in Appendix A. The joint model achieves similar performance to the category-specific models in Tab. 1 while requiring only 6 hours of training on one RTX 4090. This suggests that DoSs is not restricted to category-specific training, although scaling to broader and more diverse datasets remains future work.

## 4.3. Experimental Analysis

This section reports ablation studies with CHAIR to investigate the benefits of key designs of our DoSs, including the existence token, the 6D orientation representation, and canonical sorting strategies. The result is shown in Tab. 2.

**Existence token.** We ablate our design of existence token by comparing (i) with existance score ($e_k$ in Eq. 3) vs. (ii) without existance score. In this case, the model directly predicts the set of 128 superquadrics for each object. During training, we first shuffled the order of superquadrics in each training data. Then we padded the rest rows by copying existing superquadrics. The result indicates that the existence token is the design that contributes the most to the performance of our model. Comparing the first two rows of Tab. 2, the existence token significantly improves performance (by $34.96\%$ @ 1-NNA and $28.50\%$ @ COV). This indicates that explicitly modeling the presence or absence of valid superquadric primitives is crucial.

**Rotation representation.** We compare (i) Euler angles vs. (ii) continuous 6D rotation representation (Zhou et al., 2019), which avoids discontinuities and improves optimization stability. The result shows that adding the 6D rotation representation slightly improves the model performance (by $0.79\%$ @ 1-NNA and $1.96\%$ @ COV), which confirms that the continuous 6D representation stabilizes rotation modeling by avoiding discontinuities inherent in Euler angles. However, the performance of the 6D representation and Euler angles is comparable. We believe this may be due to the fact that, when Euler angles are constrained within a single period (e.g., from $-\pi$ to $\pi$), discontinuities in Euclidean space occur only at the boundaries, which are rarely encountered in our case.

**Canonical sorting.** In this section, we study three order-

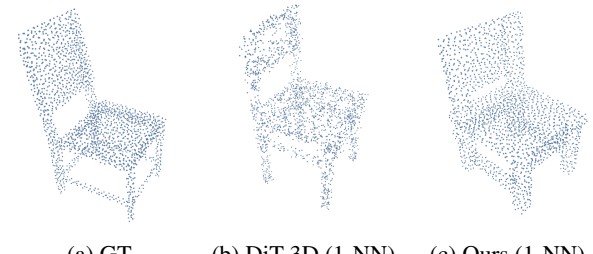

|        (a) GT        |   (b) DiT-3D (1-NN)   |   (c) Ours (1-NN)   |

*Figure 4.* Resolution free point decoding on CHAIR. We decode generated outputs into 2,048 surface points for visualization and evaluation. DoSs produces a more complete part structure than DiT-3D in this example, while CD-based distributional metrics still favor voxel diffusion spaces due to their closer match to point-level detail and sampling statistics.

ing strategies: (i) random order, (ii) sorting primitives by decreasing volume, and (iii) sorting by vertical position ($y$-axis) to align with gravity-consistent structure in chairs. As shown in the last three rows, position based sorting reduces 1-NNA from $57.57\%$ to $55.28\%$ and increases COV from $50.37\%$ to $50.93\%$. Meanwhile, volume-based sorting reduces 1-NN accuracy from $57.57\%$ to $53.80\%$ and increases coverage from $50.37\%$ to $51.13\%$. This suggests a consistent and geometrically meaningful ordering is beneficial to the model performance and sorting superquadric by volume can yield better results for the final outcome.

## 4.4. Capabilities Enabled by DoSs

We showcase three capabilities that highlight the scalability and controllability of DoSs in a compact, explicit space.

### 4.4.1. RESOLUTION FREE POINT DECODING

DoSs predicts superquadric tokens, which we decode into point clouds using the sampling pipeline in Sec. 3.4.1: we sample surface points for each predicted primitive, remove points that lie inside other primitives to handle overlaps, and apply FPS to obtain a point cloud of any requested size $N$. Fig. 4 shows an example on CHAIR (2,048 points). DiT-3D exhibits missing or deformed regions, while DoSs yields a more complete and coherent part layout. This is expected because DoSs operates in an explicit primitive space: each token represents a closed, smooth geometric part with structured degrees of freedom (shape, size, pose), which acts as a strong prior during generation.

Better visual plausibility does not necessarily imply better 1-NNA/COV under CD. These metrics depend only on nearest neighbor distances between point clouds, and CD is sensitive to point placement, density, and fine-scale details. Our decoding enforces a specific surface sampling distribution and overlap filtering, which can increase CD even when global structure looks cleaner. In addition, primitive tokens can bias samples toward smoother, more regular shapes, which may reduce fine-grained diversity and lower COV.

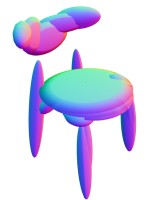 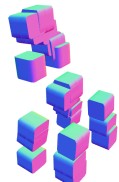 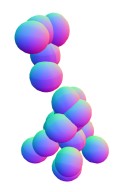

(a) $\epsilon_1 = \epsilon_2 = 1$    (b) $\epsilon_1 = \epsilon_2 = 0, \mathbf{s}_k = const$   (c) $\epsilon_1 = \epsilon_2 = 1, \mathbf{s}_k = const$

*Figure 5.* Failure modes of constrained denoising under overly restrictive constraints. Fixing exponents (a) already narrows the feasible shape family; additionally clamping all three size axes (b,c), i.e., $\mathbf{s}_k = const$, can over-constrain the process, leading to degenerate or implausible primitive assemblies.

### 4.4.2. STRUCTURE AWARE EDITABILITY

DoSs generates shape as a compact set of explicit superquadric tokens, so edits can be applied directly in token space using the operations in Sec. 3.4.2 (part deletion via the existence channel, part scaling via axis lengths, part reposing via pose parameters, and part reshaping via the shape exponents). Fig. 1 shows CHAIR edits produced by these simple parameter level manipulations. In contrast, dense diffusion spaces (points, voxels, meshes) and latent or implicit representations do not expose stable part level parameters. Achieving comparable edits typically requires additional conditioning, guidance, or post processing. Their fine grained, localized control is harder because the edit target is entangled across many coordinates or latent variables rather than a small number of interpretable part parameters.

### 4.4.3. GEOMETRIC AWARE DESIGN

DoSs supports geometric aware design by constraining interpretable token dimensions during sampling, and only denoising the remaining free variables. Following constrained denoising ideas such as DPS (Chung et al., 2022), our design interface in Sec. 3.4.3 enforces simple hard constraints via per step clamping or projection in token space. For example, to design a chair from bamboo like parts with fixed radius, we constrain each primitive to be cylinder like by setting the shape exponents to a target pair (e.g., $\epsilon_{k1} \approx 0$, $\epsilon_{k2} = 1$) and fixing the cross section $a_{k1} = a_{k2} = r$, while denoising the remaining variables (height $a_{k3}$, pose, and existence). Similarly, a plank only design is obtained by fixing a cuboid like shape and clamping one axis length as the thickness. Fig. 1 shows representative constrained generations.

This capability is difficult to realize in dense or implicit diffusion spaces, where constraints must be expressed over thousands to millions of coupled variables (voxels, points, or latents), typically requiring additional guidance or post processing, and fine grained part level control is much less direct. However, constrained denoising also has limitations. If the constraints are overly restrictive, the feasible shape family can become too small for the learned prior, causing degenerate or implausible primitive assemblies. Fig. 5

shows typical failure modes when clamping exponents and, more severely, clamping all size axes.

## 5. Limitations and Future Works

**Limitations.** Diffusion over superquadrics (DoSs) trades representational capacity for compactness and interpretability. As a result, with a bounded number of primitives, the model is biased toward piecewise-smooth, low-frequency geometry and can under-represent thin structures, sharp edges, deep concavities, and topology-heavy details that are easier to express in dense fields or meshes. In addition, DoSs inherits the failure modes of the superquadrics fitting methods: imperfect fitting introduces structured label noise (e.g., missed parts and unstable poses), while non-identifiability of parameters (multiple equivalent superquadrics parameterizations for nearly identical surfaces) makes the target distribution in token space more multimodal than in surface space, which can hinder stable learning. Finally, our evaluation relies on CD-based 1-NNA/COV computed on point clouds sampled from decoded surfaces. These Chamfer-based metrics can be misaligned with visual quality and may under-reflect our gains; more broadly, unified 3D generation metrics remain an open problem.

**Future Work.** The limitations and strengths of DoSs suggest two promising directions: improving fine geometric detail and advancing physically grounded generation. A central challenge in 3D diffusion is to reconcile object-level structure, such as part layout, symmetry, and functional organization, with surface-level detail, such as thin elements, sharp boundaries, and high-frequency geometry. DoSs deliberately emphasizes coherent global structure by diffusing a compact set of primitive parameters, but this abstraction can limit local geometric fidelity. To improve visual quality, we will explore more accurate superquadric fitting/decomposition, more expressive primitive variants, and improved parameter identifiability. We will also study hybrid pipelines where DoSs generates a structurally consistent coarse shape that is later refined by higher-capacity representations, such as local implicit patches, residual point refinement, or mesh/field upsampling.

In parallel, we will extend DoSs toward dynamic and physically grounded settings, where efficiency, feasibility, and editability are often as important as surface realism. Examples include simulation-valid generation, collision/contact reasoning, stability constraints, articulated or part-aware objects, and large-scale world modeling with long-horizon generation. DoSs is well suited to these applications: its explicit primitives make constraints such as non-penetration, support, symmetry, contact, and stability easy to impose and verify, while its compact token space enables fast sampling and iterative feasibility correction without the heavy cost of dense 3D diffusion.

# 6. Conclusion

We presented DoSs, which shifts diffusion from dense geometric representations to a compact and explicit space of superquadric tokens. By denoising explicit primitive parameters (pose, scale, and shape) rather than thousands-to-millions of dense values, DoSs substantially reduces the diffusion-state dimensionality while retaining the ability to decode 3D surfaces and point clouds. Across standard benchmarks, DoSs achieves competitive surface fidelity and distributional quality after decoding to point clouds, while offering shorter training and inference times. Moreover, DoSs enables structure-aware capabilities that are difficult to realize in dense diffusion spaces, including resolution-free sampling, part-level editability, and constrained denoising for geometry-aware design. Overall, DoSs suggests that diffusing in an explicit and compact primitive space is a practical route toward scalable and controllable 3D generation. Future work includes improving tokenization and denoisers for higher fidelity, most promisingly, leveraging DoSs' compact and explicit tokens for efficient multimodal/scene conditioning and for physically grounded, temporally structured generation with geometric constraints.

# Impact Statement

This paper aims to advance 3D generative modeling by moving the diffusion process from dense geometric representations to compact and explicit superquadric tokens. By reducing the dimensionality of the generative space, our approach can lower compute and memory costs of training and sampling, which may reduce energy use and improve accessibility of 3D generation research and applications for groups with limited hardware. The explicit, part-based parameterization can also support more interpretable and controllable 3D synthesis, which may benefit downstream uses such as simulation, robotics, AR/VR content creation, and rapid prototyping.

As with many general-purpose 3D generation techniques, there are potential risks. Generated shapes could be repurposed to aid the design of harmful or unsafe objects, or be used in contexts where fabricated parts require strict safety certification. Moreover, training data may encode biases in object categories or styles, leading to uneven performance or stereotyped outputs; and generated shapes may inadvertently resemble copyrighted or proprietary designs if such patterns exist in the data. To mitigate these concerns, we encourage responsible release and use practices (e.g., adherence to dataset and model licenses, clear documentation of intended use and limitations, and optional filtering of sensitive categories where applicable). We also emphasize that outputs from this model are not validated for safety-critical deployment and should undergo application-specific review, testing, and compliance checks before real-world use.

# Acknowledgement

We thank anonymous ICML reviewers for their useful feedback and suggestions.

This research was supported in part by the Advanced Robotics Centre (ARC) of the National University of Singapore.

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

## A. Results on Multi-class Training

*Table 3.* Quantitative evaluation on different object categories.

| Category | 1-NNA-CD | COV-CD |
|----------|----------|--------|
| Chair    | 55.70    | 50.19  |
| Airplane | 60.20    | 45.36  |
| Car      | 61.11    | 47.78  |

We also trained DoSs in CHAIR, CAR, AIRPLANE together to show the effectiveness of our model on multi-class training. Since our model is unconditional and generating objects of a specific type requires taking the category information as condition, we modified our model by encoding the category index with 3-layer MLP, concatenating with time embedding and input to the noise prediction network in DDPM. We evaluated the multi-class model with 1000 objects of each category, the result of 1-NNA and COV is illustrated in Tab. 3. We can observe that the DoSs achieves competitive generation results against category-specific models for all metrics. This result demonstrates that DoSs hold the potential to simultaneously generate objects of all categories by training a single global model. The multi-categories model is trained with around 8,000 objects for 6 hours on one RTX 4090.

## B. Primitive Budget Analysis

DoSs represents each shape using a fixed token budget of $K_{\max} = 128$ superquadric slots, while the actual number of active primitives varies across objects and is controlled by the existence score. To justify this choice, we report the statistics of the superquadric decompositions used in our ShapeNet experiments in Tab. 4.

*Table 4.* Statistics of the number of superquadrics per shape in the extracted ShapeNet superquadrics. P90 denotes the 90th percentile: 90% of shapes in that category use no more than this number of primitives.

| Category | Mean  | Median | P90 | Max |
|----------|-------|--------|-----|-----|
| AIRPLANE | 12.98 | 13     | 17  | 52  |
| CHAIR    | 21.08 | 19     | 36  | 123 |
| CAR      | 18.64 | 13     | 38  | 100 |

The statistics show that $K_{\max} = 128$ is sufficient for the current ShapeNet setting. All observed decompositions fit within this budget, with maximum primitive counts of 52, 123, and 100 for AIRPLANE, CHAIR, and CAR, respectively. Moreover, the P90 values are much smaller than 128, indicating that most shapes use only a small fraction of the available slots. This suggests that, for the evaluated categories, the fixed primitive budget is not the main bottleneck.

Increasing the budget to larger states such as $256 \times 15$ or $512 \times 15$ may improve representational capacity for more complex objects, but it would also increase the diffusion-state size and sampling cost. For the current ShapeNet benchmarks, we therefore use $K_{\max} = 128$ as a compact budget that covers the extracted primitives while keeping the denoising state small. Exploring larger or flexible-length primitive generation, such as token-by-token autoregressive superquadric generation, is an interesting direction for future work on more diverse datasets such as Objaverse.

## C. Parameter Sensitivity Analysis

DoSs denoises heterogeneous superquadric parameters, including position, rotation, size, shape exponents, and existence scores. These parameters have different physical meanings and may affect the denoising process differently. To better understand this behavior, we conduct a perturbation study by perturbing one parameter group at a time during sampling.

For a fixed sampling seed, we first generate a reference output $\tilde{Z}_0$. We then repeat sampling with the same seed, but add random perturbation to only one parameter group while keeping the remaining groups unchanged. Specifically, for each group $g \in \{\text{shape}, \text{size}, \text{rotation}, \text{position}, \text{existence}\}$, we add $10\%$ random noise to the corresponding normalized token dimensions during denoising. After sampling, we compare the perturbed output $\tilde{Z}_0^{(g)}$ with the reference output using the Manhattan distance: $\Delta_g = \left\| \tilde{Z}_0^{(g)} - \tilde{Z}_0 \right\|_1$. Since different parameter groups have different dimensionalities, we also report the deviation per dimension, computed as $\Delta_g / d_g$, where $d_g$ is the dimensionality of the perturbed group.

*Table 5.* Parameter sensitivity under group-wise perturbations during denoising. For each parameter group, we add 10% random noise to that group only and measure the final deviation from the unperturbed output using Manhattan distance. The per-dimension deviation accounts for different group dimensionalities.

| Parameter group | Dim. | Total deviation | Deviation / dim. |
|---|---|---|---|
| Shape | 2 | 153.90 | 76.95 |
| Size | 3 | 104.20 | 34.73 |
| Rotation | 6 | 138.78 | 23.13 |
| Position | 3 | 132.40 | 44.13 |
| Existence | 1 | 101.59 | 101.59 |

The results suggest that the denoising process is most sensitive to coarse structural factors. In particular, the existence score has the largest per-dimension deviation, which is consistent with its role in determining whether a primitive is active. Shape parameters are also highly influential, as they control the primitive family and therefore strongly affect the global geometry. Position perturbations produce a non-negligible deviation because they directly change the spatial layout of parts. In contrast, size and rotation have smaller per-dimension deviations in this diagnostic study.

This analysis should be interpreted as a preliminary robustness diagnostic rather than a complete perceptual evaluation. The reported distances are measured in the normalized token space, not directly in surface space. Nevertheless, the trend supports the importance of explicitly modeling existence and shape parameters, and helps explain why the existence-token design contributes strongly to the performance observed in Tab. 2.

## D. Discussion

**Scaling diffusion in 3D: where does the compute really go?** A central challenge in 3D diffusion is that iterative denoising amplifies the cost of the chosen representation: even modest increases in spatial resolution or point count can multiply per-step compute/memory, and the cumulative cost over dozens to hundreds of steps becomes the bottleneck. DoSs directly targets this challenge by moving diffusion to a compact, continuous token space, making long denoising chains and repeated sampling (e.g., for selection, reranking, or constrained generation) substantially more practical. This reframes a common question in 3D diffusion: *how to afford iterative refinement in high-dimensional spaces*, into a representation choice: reducing state dimensionality can unlock additional algorithmic degrees of freedom (e.g., stronger conditioning, constraint projection, or multi-try sampling) that are otherwise prohibitively expensive.

**Controllability and constraints: from "prompting" to verifiable geometry.** A major limitation of many 3D diffusion systems is that control is often indirect: users condition on text or images but lack an explicit interface to enforce geometric properties. DoSs offers a different control primitive: constraints can be expressed directly on interpretable parameters (pose/scale/shape) and checked against decoded geometry (e.g., collision, support, symmetry). This suggests a broader shift in how we think about controllable 3D diffusion: rather than relying solely on learned conditioning signals, we can incorporate *explicit constraint operators* during sampling. The open challenge is to make such constraints robust (avoid over-constraining and mode collapse) while remaining expressive (support multiple feasible solutions); compact token spaces make iterative constraint enforcement and multi-hypothesis sampling substantially more feasible.

**Evaluation: decoding bias, metric sensitivity, and the open problem of unified benchmarks.** While point-cloud metrics such as CD/EMD-based 1-NNA and COV are widely used, they entangle the generative model with the *decoder and sampling procedure*. For token-based methods, small changes in surface sampling (density, stratification, noise, coverage of thin structures) can noticeably affect distance estimates, and thus change 1-NNA/COV even when rendered geometry looks similar. Moreover, distributional metrics can disagree with perceptual quality: a method may generate visually plausible shapes that are nonetheless "easy to separate" under nearest-neighbor tests if it induces systematic biases (e.g., smoother surfaces, reduced micro-variations, or a narrower manifold after decoding). These issues highlight an open problem for the community: establishing *unified, representation-agnostic evaluation* that remains stable across decoding pipelines and captures both structural plausibility and geometric fidelity. A unified metric that is simultaneously faithful, stable across representations, and aligned with human judgment remains an important open question in 3D generation.

## E. Qualitative Visualization

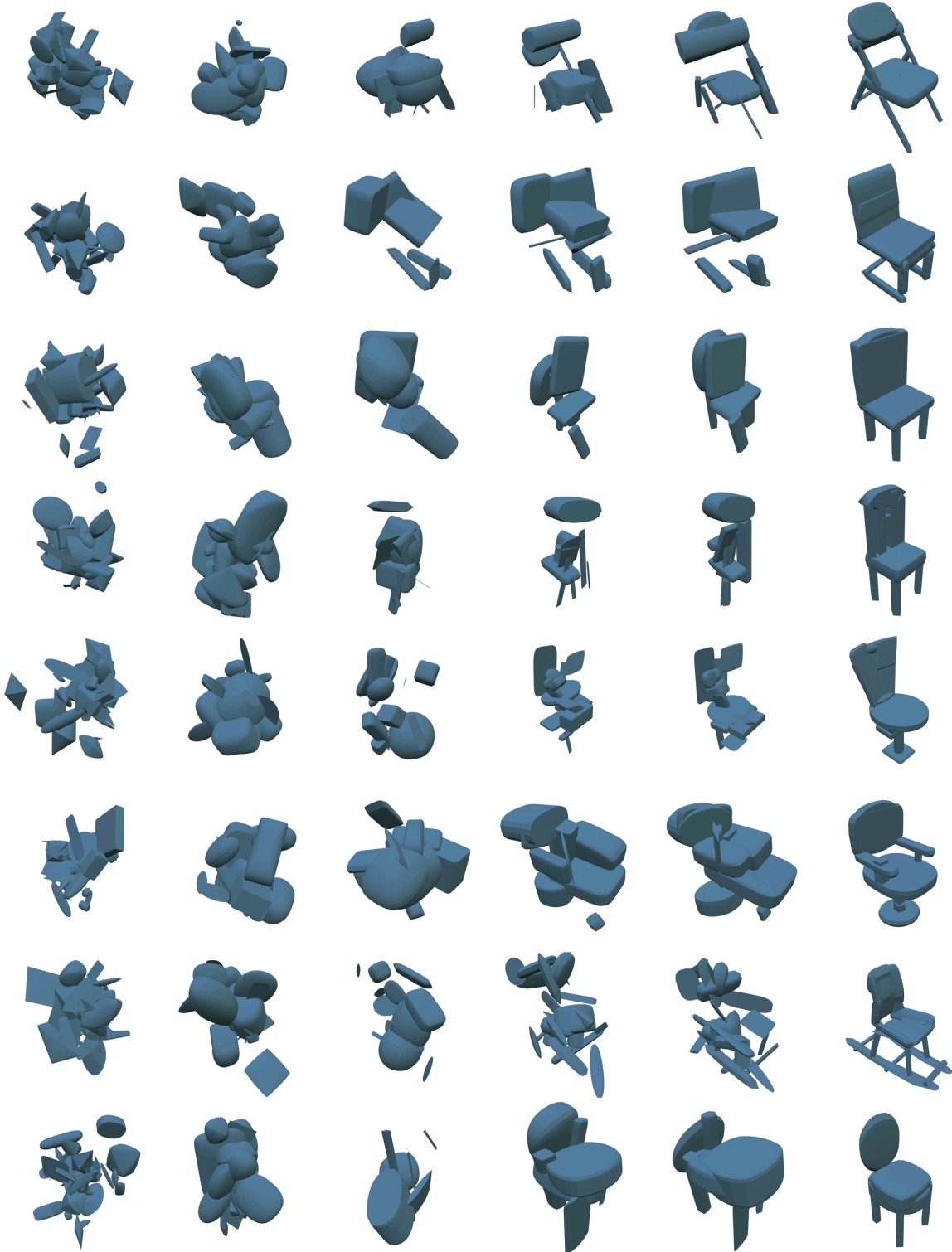

*Figure 6.* Qualitative visualizations of the denoising process on CHAIR shape generation. The results of generating from random noise to final 3D shapes are shown in left-to-right order.

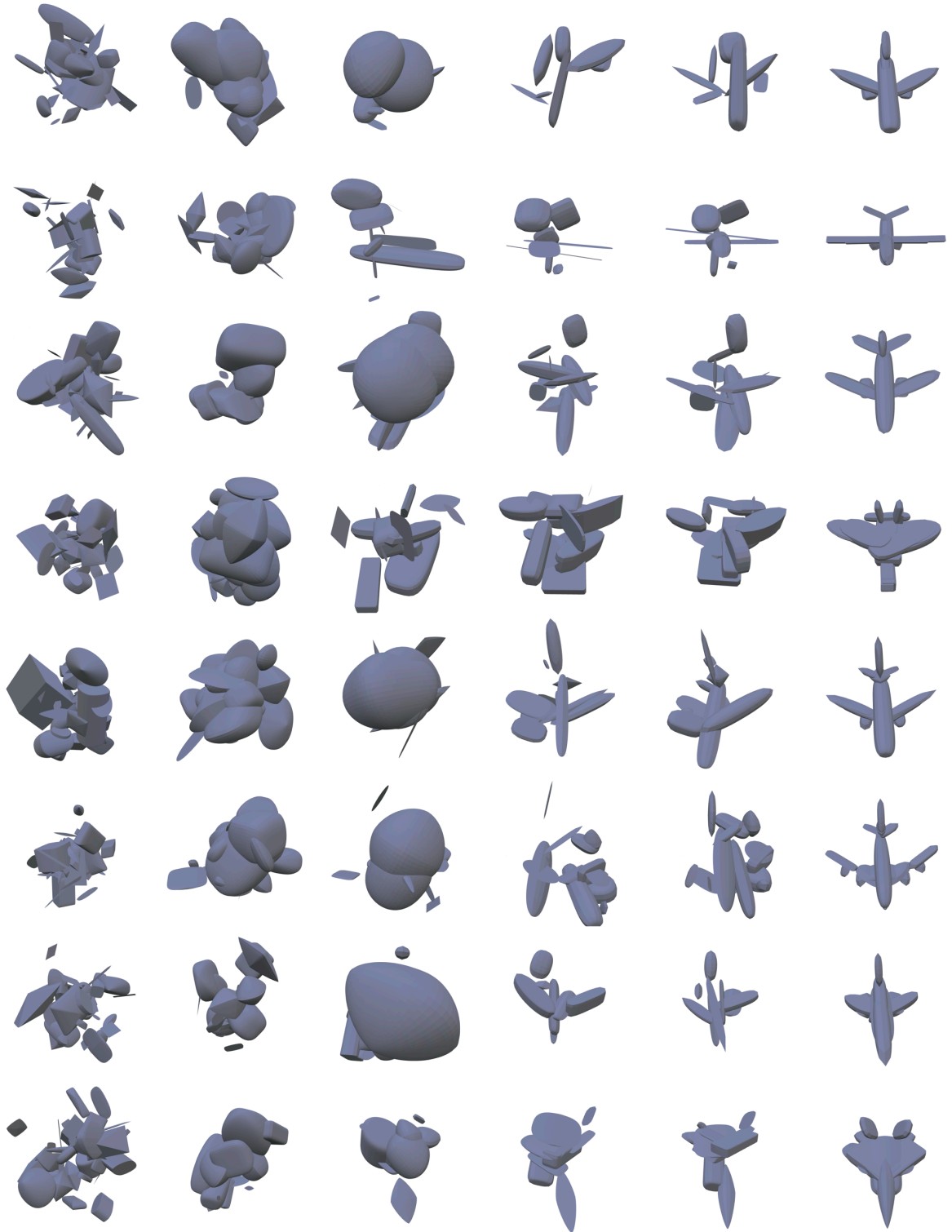

*Figure 7.* Qualitative visualizations of the denoising process on AIRPLANE shape generation. The results of generating from random noise to final 3D shapes are shown in left-to-right order.

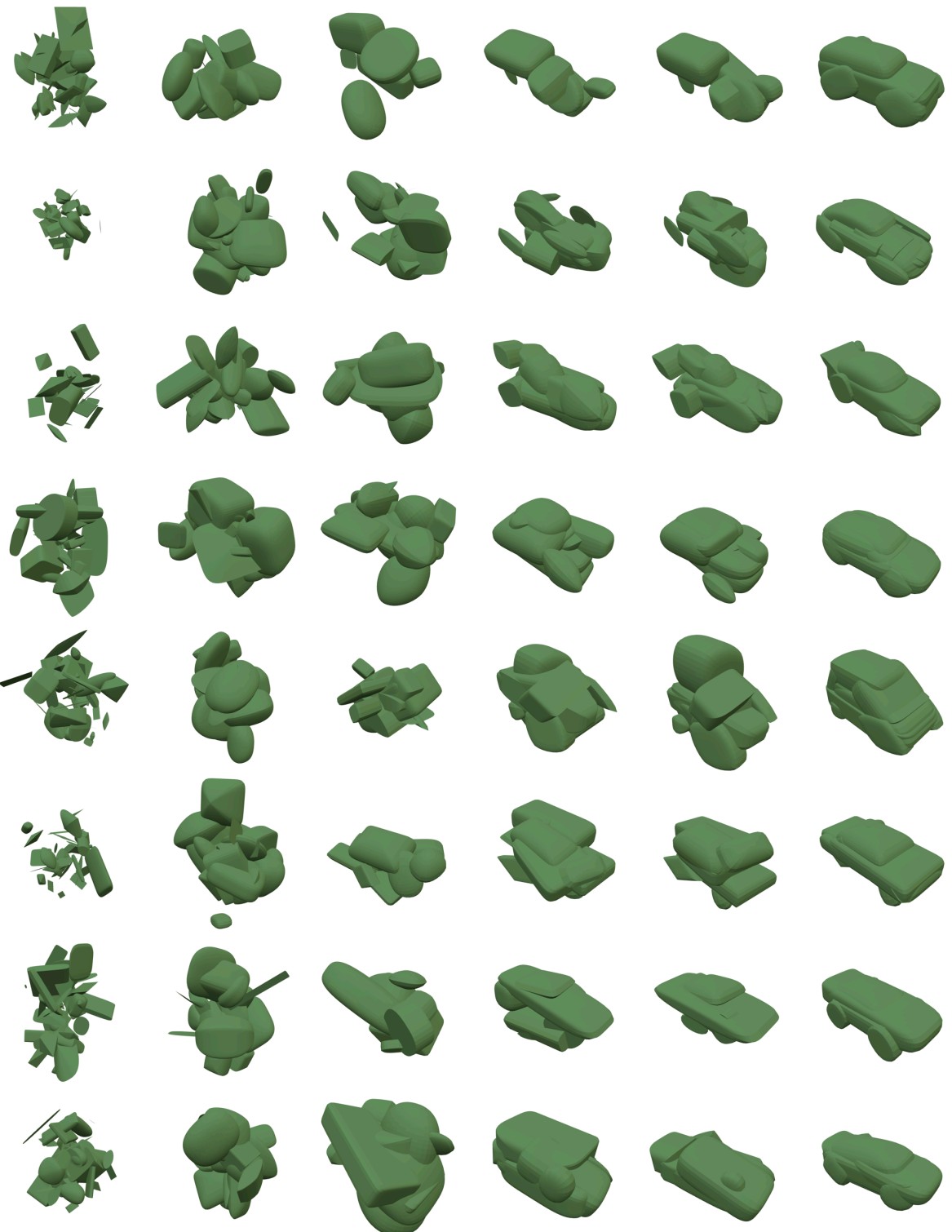

*Figure 8.* Qualitative visualizations of the denoising process on CAR shape generation. The results of generating from random noise to final 3D shapes are shown in left-to-right order.

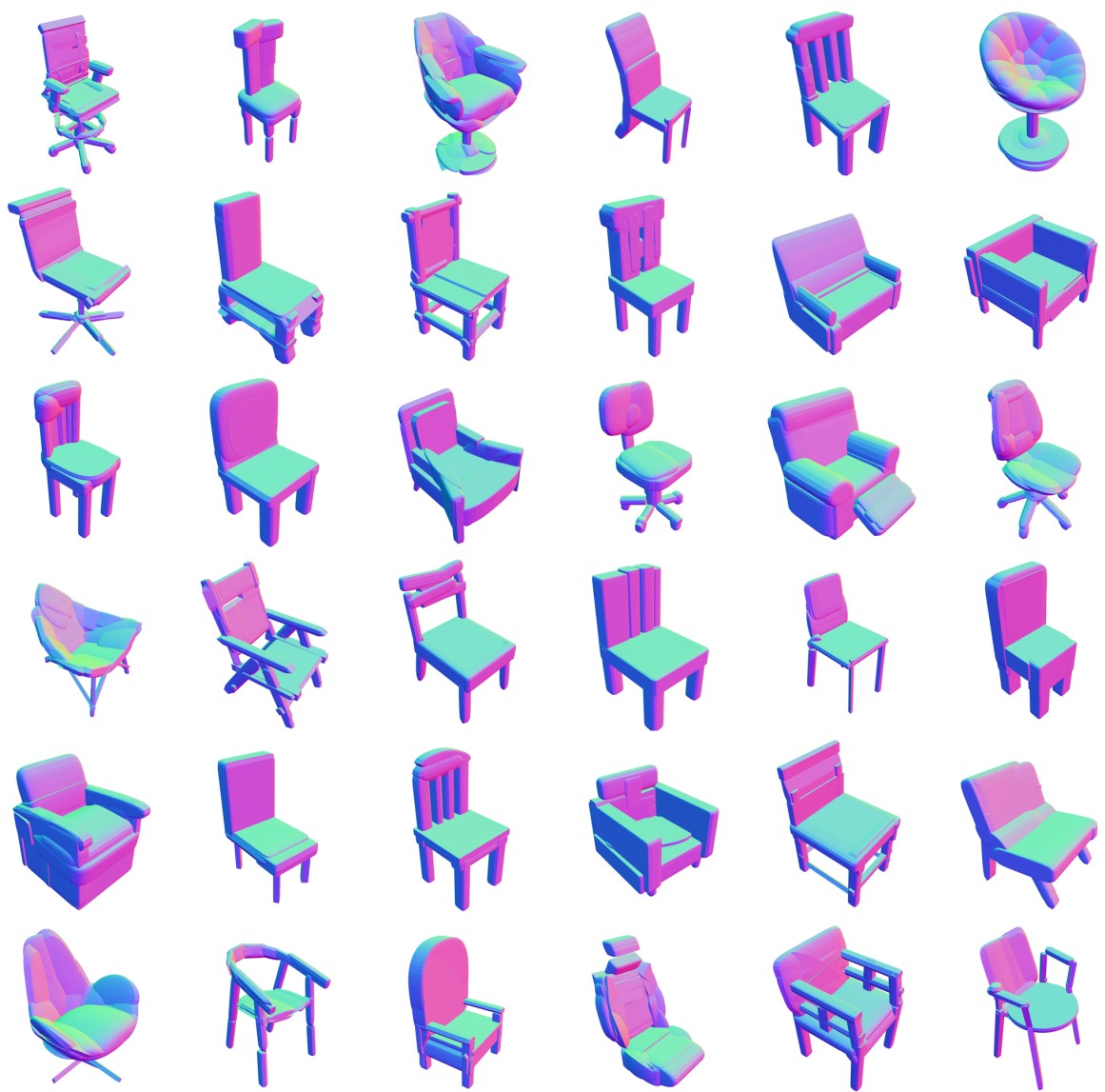

*Figure 9.* Qualitative visualizations of high-fidelity and diverse results on CHAIR shape generation.

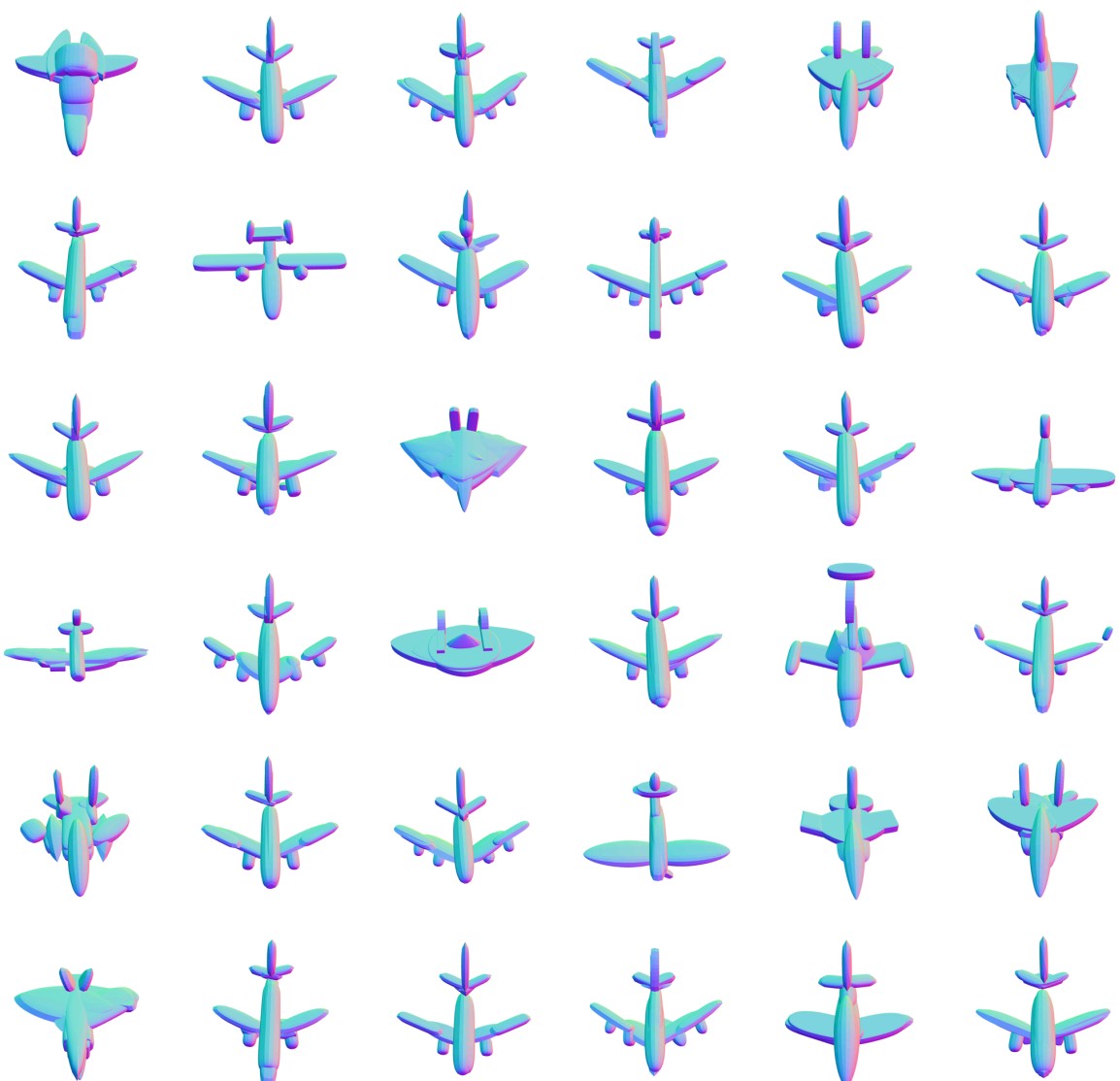

*Figure 10.* Qualitative visualizations of high-fidelity and diverse results on AIRPLANE shape generation.

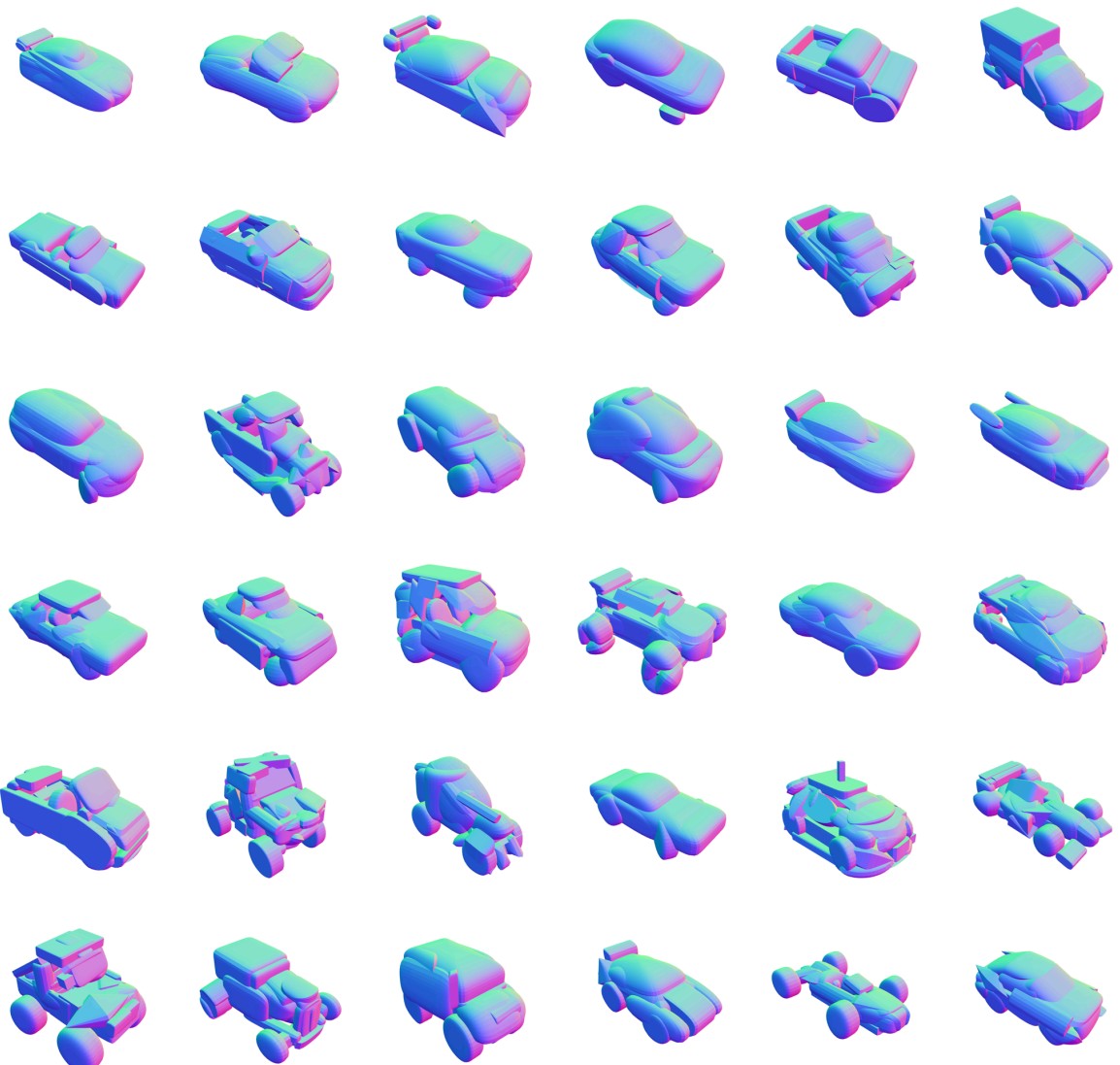

*Figure 11.* Qualitative visualizations of high-fidelity and diverse results on CAR shape generation.

