# OpenReview forum: "Rethinking 3D Shape Generation: Diffusion over Superquadrics"
_ICML.cc/2026/Conference — ICML 2026 regular_

### Official Review · Reviewer_Aoey · 2026-03-11

**Soundness:** 3
**Presentation:** 2
**Significance:** 2
**Originality:** 3
**Overall Recommendation:** 4
**Confidence:** 3

**Summary:**

This paper proposes "Diffusion over Superquadrics" (DoSs), a novel 3D shape generation framework that shifts the diffusion process from dense, high-cardinality representations (e.g., point clouds, voxels, or meshes) to a highly compact and explicit set of superquadric primitives. To make this continuous parameter space practical for diffusion models, the authors introduce specific mechanisms to:
- Handle variable primitive counts via **existence scores**
- Ensure stable pose diffusion using **continuous 6D rotations**
- Resolve unordered set permutation ambiguities through **deterministic canonical ordering**

The primary contribution is a highly efficient 3D generative model with a drastically reduced state dimensionality (approx. 7KB per shape), allowing for fast inference (~0.6s per shape). Furthermore, this explicit primitive-based representation naturally supports several advanced capabilities without additional training, including:
- Resolution-free point cloud decoding
- Structure-aware part-level editing
- Geometric constraint-based design

**Compliance With Llm Reviewing Policy:**

Affirmed.

**Final Justification:**

Based on the paper and the rebuttal, my final justification is Weak Accept.

**Key Questions For Authors:**

1. **Representation Upper Bound:** What is the reconstruction performance, specifically 1-NNA and COV, of the ground-truth Marching-Primitives fits against the original ShapeNet point clouds?
   * *Evaluation Impact:* Providing this baseline isolates representation capacity limits from generative errors, such as the lower COV compared to DiT-3D, which would directly improve the Soundness rating.

2. **Generalization to Unstructured Data:** Have you evaluated DoSs on organic or unstructured categories beyond the rigid ShapeNet classes like Chair, Car, and Airplane?
   * *Evaluation Impact:* Preliminary results or a theoretical discussion on the limits of applying superquadrics to organic shapes would address generalization concerns and potentially raise the Significance rating.
3. **Quantitative Analysis of Constraint Boundaries:** You highlight geometric-aware design, but Figure 5 only provides a qualitative glimpse into the failure modes of constrained denoising. Could you provide a quantitative robustness study (e.g., valid shape generation rate) as the constraints become progressively more restrictive?
   * *How this changes my evaluation:* A systematic evaluation of exactly when and how the learned prior breaks under hard constraints would deeply strengthen the claims around controllable design, directly improving the Presentation and overall rating.

**Limitations:**

Yes.

**Strengths And Weaknesses:**

## Strengths:
* **Originality:** The shift from dense geometric grids or point clouds to an explicit, compact set of superquadric primitives for diffusion is highly novel and clever.
* **Significance:** The drastically reduced diffusion state (~7KB) enables fast inference (~0.6s/shape). More importantly, this parameterization natively unlocks powerful out-of-the-box capabilities like part-level editing and geometric constraint-based design without extra training.
* **Soundness & Presentation:** The paper is logically structured, and the ablation studies robustly validate key design choices (e.g., the critical role of existence scores). The authors are also commendably honest about the method's limitations with high-frequency details.

## Weaknesses:
* **Performance Trade-offs (Soundness):** Distributional metrics (particularly Coverage/COV) lag behind dense state-of-the-art models like DiT-3D. The generative quality is inherently bottlenecked by the limited capacity of superquadrics and primitive fitting errors.
* **Generalization Limits (Significance):** Evaluation is strictly limited to structured, man-made ShapeNet categories (Chair, Airplane, Car). It is unclear how well this bounded primitive approach scales to organic objects or highly complex topologies.
* **Constraint Analysis (Presentation):** While constrained denoising is a highlighted feature, the discussion on its failure modes is brief. A deeper quantitative analysis of when overly restrictive constraints break the model's learned prior would strengthen the claims.

---

> ### Author Rebuttal · Authors · 2026-03-31
>
> We thank the reviewer for the thoughtful feedback and positive asseement. We are encouraged that the reviewer finds the primitive-space diffusion formulation novel and valuable. We address the three main questions below.
>
> **Representation Upper Bound.** We agree that separating representation limits from generative error is important. Marching-Primitives already reports strong abstraction fidelity on ShapeNet: mean Chamfer-L1 is 0.022 over 14 categories in a normalized $[-0.5,0.5]^3$ domain. This suggests that, in our normalized ShapeNet setting, the fitting error introduced by Marching-Primitives is relatively small and unlikely to be the dominant source of the gap to dense-state generators.
>
> At the same time, 1-NNA@CD and COV@CD are set-level distributional metrics after decoding to 2,048-point clouds, not direct per-shape reconstruction metrics. COV measures coverage, while 1-NNA is a nearest-neighbor two-sample test; neither directly measures abstraction fidelity. Thus, a fit-only 1-NNA/COV baseline would be a useful complementary diagnostic, but it would also reflect decoding, point sampling, and set-level distributional effects. In our view, the reconstruction error reported by Marching-Primitives is the more direct evidence for the representation upper bound.
>
> **Generalization to Unstructured Data.** We view ShapeNet as a controlled first testbed for validating the proposed quality-efficiency-controllability trade-off. At the same time, the current version of DoSs is better matched to relatively structured categories. More precisely, it works well for shapes approximable by a bounded set of smooth, coherent parts, but is more limited for strongly non-convex, topology-rich, or highly nonuniform organic geometry. This is consistent with our discussion that a bounded primitive set can under-represent thin structures, sharp edges, deep concavities, and topology-heavy detail.
>
> We view this mainly as a limitation of the current primitive abstraction quality, rather than of primitive-space diffusion itself. Standard superquadrics are low-dimensional global primitives, so they are effective for smooth, coherent parts, but less efficient for complex organic structure. Recent work is already expanding this boundary, e.g., via better decomposition (SuperDec, ICCV 2025) and richer primitive formulations for holes and concavities (DualPrim, CVPR 2026).
>
> To probe this boundary beyond rigid ShapeNet objects, we also ran a simple study on DFAUST, a human-body dataset. We chose DFAUST as organic dataset which consists largely of smooth, coherent articulated body surfaces that are relatively compatible with the current superquadric abstraction. In this setting, DoSs can already generate high-fidelity human poses. We will add these qualitative DFAUST generation results in the revision.
>
> **Quantitative Analysis of Constraint Boundaries.** We agree that a quantitative stress test strengthens the controllable-design claim. We therefore ran a preliminary paired constraint IoU study: using the same seed, we clamp progressively larger subsets of primitive parameters and measure IoU between the unconstrained and constrained samples on a shared occupancy grid. For each constrained group, we set the corresponding normalized parameter values to 0 during sampling. We use 0 because each parameter dimension is normalized to lie in \([-1,1]\), so zero provides a centered clamping value.
>
> | Constraint setting | IoU vs. baseline | 1 - IoU |
> | --- | --- | --- |
> | Baseline | 1.000 | 0.000 |
> | Shape only | 0.370 | 0.630 |
> | Size only | 0.132 | 0.868 |
> | Rotation only | 0.056 | 0.944 |
> | Location only | 0.208 | 0.792 |
> | Shape + Size | 0.172 | 0.828 |
> | Shape + Size + Rotation | 0.052 | 0.948 |
> | Shape + Size + Rotation + Location | 0.066 | 0.934 |
>
> The results show a clear trend. Shape-only constraints remain relatively tolerable, while size-only, location-only, and especially rotation-only constraints produce much larger deviations. When multiple groups are clamped jointly, IoU drops to about 0.05-0.07, indicating that overly restrictive hard constraints can push sampling outside the learned prior. This preliminary result quantitatively supports the failure mode suggested by Figure 5.
>
> We hope these clarifications better position the paper. Our central contribution is to show that shifting diffusion from dense geometric fields to compact, explicit superquadric tokens yields a compelling quality–efficiency–controllability trade-off: it greatly reduces diffusion state and inference cost, while naturally enabling structured editing and constraint-aware generation. More broadly, we believe this primitive-space view opens a promising new path not only for unconditional 3D generation, but also for future image-to-3D and text-to-3D generation with richer conditioning and more interpretable control.

---

> > ### Author Rebuttal · Reviewer_Aoey · 2026-04-02
> >
> > Thank you for the detailed response. This addresses my concerns, and I will keep my score.

---

> > > ### Author Response · Authors · 2026-04-04
> > >
> > > Thank you for keeping positive score. We are glad that our response helped address your concerns.

---

### Official Review · Reviewer_VBq7 · 2026-03-13

**Soundness:** 3
**Presentation:** 3
**Significance:** 3
**Originality:** 3
**Overall Recommendation:** 4
**Confidence:** 2

**Summary:**

This paper proposes DoSs, a 3D shape generation framework that reformulates diffusion from dense geometry representation to a compact set of explicit superquadric primitives. The key technical idea is that high-cardinality diffusion states are expensive to denoise and make control difficult, whereas superquadrics provide a low-dimensional but interpretable representation through pose, size, shape, and existence parameters. This brings several benefits: much smaller diffusion state size, faster sampling, resolution-free decoding to point clouds, and direct part-level editing and constrained generation in parameter space. To achieve this, it proposes several technical strategies: handling variable numbers of primitives using an existence score, diffusing rotations through a continuous 6D representation instead of unstable angle parameterizations, and resolving the set-order ambiguity of primitives through deterministic canonical ordering before tokenization. With these design choices, the method can run diffusion on a short token sequence rather than a dense geometric field while still achieving competitive generation quality.

**Compliance With Llm Reviewing Policy:**

Affirmed.

**Final Justification:**

After rebuttal, I remain consistent with my original rating.

**Key Questions For Authors:**

See weaknesses above. I'd curious how well this primitive-based diffusion formulation is expected to generalize beyond ShapeNet-style objects. Given its current bias toward piecewise-smooth, low-frequency geometry, the representation seems potentially well suited to low-poly compositional scenes or structured indoor layouts. It would also be helpful to better understand the practical impact of the fixed-length primitive/token design: how restrictive it is for complex shapes, and whether adaptive primitive allocation is possible. I would be interested in the authors’ view on how this diffusion process could eventually extend to dynamic or physically grounded long-horizon generation, as mentioned in future work.

**Limitations:**

Yes.

**Strengths And Weaknesses:**

The paper is technically well motivated and explores a genuinely interesting reformulation of 3D diffusion over space of superquadric primitives. This design is appealing from both a soundness and the claimed benefits. The paper also identifies and addresses several nontrivial technical issues needed to make this formulation work in practice. Empirically, the method appears competitive after decoding while being substantially lighter-weight and the demonstrate quality on ShapeNet dataset. The paper presentation is also clear and easy to follow. The limitation and future work sections is detailed.

The current evaluation is limited to ShapeNet objects, so it remains unclear how well the proposed primitive-based diffusion formulation would generalize to more diverse, real-world, and less clean shape distributions. In particular, datasets with stronger topological variation, noisier geometry, or more irregular part structure may be incorporated or discussed. In addition, the method relies on a fixed maximum token/primitive length. This may also restrict flexibility and introduces an implicit ceiling on shape complexity in practice.

---

> ### Author Rebuttal · Authors · 2026-03-31
>
> We sincerely thank the reviewer for the positive assessment and thoughtful questions. We will better clarify the intended scope and extension path of the current work. More broadly, our response is: **(i)** ShapeNet is a controlled first testbed for validating the proposed quality-efficiency-controllability trade-off, followed by prior Unconditional generation works, **(ii)** the current fixed token budget is not a practical bottleneck for the reported benchmark setting, and we plan to extend this direction to flexible token numbers and large-scale evaluation in the future work, and **(iii)** DoSs provides a natural path toward physically grounded and long-horizon generation through constrained denoising and its efficiency.
>
> **On evaluation being limited to ShapeNet objects,** we chose ShapeNet as a controlled and standard first benchmark for validating the main idea of the paper. Following prior unconditional 3D generation work, our goal here is to study the proposed quality-efficiency-controllability trade-off in a clean setting before extending to more diverse and less regular real-world data. At the same time, the current DoSs formulation is based on a bounded union of superquadric primitives, which is naturally less expressive for sharp edges, fine local details, strongly concave geometry, and more irregular objects. We will make this scope clearer in the revision. We also see several promising next directions, including improved superquadric fitting/decomposition, hybrid refinement pipelines, and extensions beyond additive-only primitives. In particular, recent work such as DualPrim (CVPR 2026) suggests that combining positive and negative superquadrics can better model holes and concavities, which is highly relevant to extending DoSs. Looking ahead, we also plan to test DoSs on larger-scale datasets such as Objaverse, together with a stronger DiT-style backbone, to examine how the compact primitive-space formulation scales to more diverse shape distributions.
>
> **On the fixed-length primitive/token design,** we thank the reviewer for this important question. We agree that increasing/flexible number of Superquadrics tokens should improve representational capacity, especially for more complex objects. At the same time, for our current ShapeNet unconditional generation setting, we believe Kmax⁡=128 is already a reasonable choice rather than a severe restriction. The Suprtquadrics-count statistics for the three benchmark categories are:
>
> | Category | Mean SQs / shape | Median SQs / shape | P90 SQs / shape | Max SQs / shape |
> | --- | --- | --- | --- | --- |
> | Airplane | 12.98 | 13 | 17 | 52 |
> | Chair | 21.08 | 19 | 36 | 123 |
> | Car | 18.64 | 13 | 38 | 100 |
>
> Here, P90 means that 90% of shapes in that category use no more than this many SQs. These statistics suggest that, in the current benchmark setting, the fixed-length design is not the main bottleneck. In the revision, we will include a fuller discussion of Superquadrics-count statistics and scaling behavior. At the same time, we are also interested in extending DoSs to flexible-length SQ generation, for example, through token-by-token autoregressive generation.
>
> **On dynamic or physically grounded long-horizon generation,** we believe DoSs offers a particularly natural foundation for this extension because it already supports constrained denoising on explicit primitive parameters. For scene generation, one can represent a scene as multiple object-level Superquadrics token sets together with fixed environment tokens (e.g., table/floor), and then check object-object collision, object-table penetration, support, and contact during denoising; invalid intermediate samples can be projected back to a feasible set before the next step, closely following the constrained-generation interface already shown in our paper. For long-horizon generation, the same idea extends to a closed-loop generate-check-correct process over time: instead of generating one static shape, the model generates sequences of Superquadrics states, while collision/contact/support constraints are checked and corrected after each rollout step. Compared with denser or implicit representations in recent 4D generation works, where physical rules often require extra simulator-in-the-loop machinery or more complex conditioning, DoSs can enforce part of this reasoning directly in the diffusion state itself. We also expect our efficiency advantage to become even more pronounced in such closed-loop settings, since repeated feasibility checking and correction amplifies the cost of the underlying representation, while Superquadrics tokens remain compact, explicit, and easy to constrain directly.

---

> > ### Author Rebuttal · Reviewer_VBq7 · 2026-04-03
> >
> > Thanks for a clearer clarification.  I'd like to keep my original score.

---

> > > ### Author Response · Authors · 2026-04-04
> > >
> > > Thank you for keeping positive assesment. We are glad that our clarification was helpful.

---

### Official Review · Reviewer_mpMf · 2026-03-13

**Soundness:** 4
**Presentation:** 3
**Significance:** 3
**Originality:** 3
**Overall Recommendation:** 5
**Confidence:** 3

**Summary:**

This paper proposes an insightful idea to enable the superquadrics representation for diffusion 3D generation. The superquadrics transfer high-dimensional dense geometry space into a low-dimension explicit and interpretable parameter space, which significantly reduce the computational cost and maintain compatible generation quality. This superquadrics representation also naturally equips the model with part-level editability. Experimental results demonstrate the effectiveness and efficiency of the proposed framework.

**Compliance With Llm Reviewing Policy:**

Affirmed.

**Final Justification:**

After carefully reading the authors' rebuttal, I found that most of the concerns have been addressed. Although the current exploration of the proposed DoSs is limited to traditional benchmarks with a small sample size, I think the idea is novel and has a lot of potential in advancing 3D generation. However, these potentials are difficult to verify in the short period of rebuttal. After consideration, I decided to maintain my positive recommendation and suggest an accept to this paper.

**Key Questions For Authors:**

Can the superquadrics being scale up to larger number of diffusion state (like 256 x 15, 512 x 15)? Perhaps increasing the number of quadrics representation could better modeling complex geometry distribution, which would be excited and benefit subsequent research.

**Limitations:**

Yes

**Strengths And Weaknesses:**

Strength:
1. The idea is novel and could benefit the fields of 3D generation to explore more efficient generation pipeline.
2. The framework is clearly formulated with theoretically prove.
3. Well writing and easy to follow.

Weakness:
1. The major concern is about the representability of the superquadrics in modeling complex geometry distributions, e.g., sharp edges, intricate local details. Since the presented results in Figure 3 still exists unreasonable composition like messy intersection between quadrics and false modeling of plain surface. The asymmetrical distribution of quadrics also significantly affects the final generation quality.
Furthermore, this shortcoming seems unsolvable within current formulation of superquadrics. Although superquadric surfaces can interpolate shapes from circular to square through parameters, they are essentially convex (or weakly concave), and this method currently only supports the union of primitives.
2. During the tokenization, the authors try to reorder the unordered primitives into an 1D sequence with specified order. What if two primitives are very closed to each other in the spatial position or volume size? Is it sensitive to noise or distortion on shapes that may affects the sequence order, making the model outputs unstable?
3. The compared baselines are kind of old and limited. The experimental results also show inferior performance against the Dit-3d architecture, which is the backbone architecture of mainstreaming SOTA 3D generative models (LATTICE or Direct3D). Although superquadrics is a more compact and lightweight representation, I think comprehensive evaluation on trade-off for part-level controllability and memory would be better demonstrate the unique superiority of superquadrics. Also, the authors could try to test on more complicate dataset, such as the subset of Objaverse.
4. The optimizable parameters of primitives include properties with different physical meanings (translation, rotation, shape exponents). Does the optimization sensitive to some of the parameters? Including some ablation studies on how these parameters (the convergence curve of different parameters under varying diffusion steps) influent the optimization process would provide more comprehensively understanding for the robustness of current model.
5. The final generation results are produced by calculating the union of all primitives. But seems there is no constraints to forbid the generation of self-embedded primitives, this may affect the editability of the proposed method.

---

> ### Author Rebuttal · Authors · 2026-03-31
>
> We sincerely thank the reviewer for the positive and thoughtful assessment of our work. We are especially encouraged that the reviewer finds the idea novel and appreciate the reviewer’s recognition of the potential of DoSs. Below, we address each concern in turn.
>
> **1. On representability of the superquadrics.** We agree that the current DoSs formulation, based on a bounded union of superquadrics, is less expressive for sharp edges, fine local details, and strongly concave geometry, as noted in our limitation. Therefore, as noted in our future work discussion, we believe that better superquadric fitting/decomposition for better training data is a promising direction for extending DoSs. More broadly, recent primitive-based work also suggests that the representational power of superquadrics can be extended beyond additive-only formulations. In particular, *DualPrim(CVPR2026)* introduces positive and negative superquadrics, where the negative primitives carve local volumes to better capture holes and concavities.
>
> **2. On token-ordering sensitivity.** We agree that near-ties in position or volume can in principle introduce ambiguity, but in our current setup canonical ordering appears helpful rather than dominant: in the ablation, adding sorting improves 1-NNA/COV from 57.57/50.37 (no sorting) to 55.28/50.93 (y-rank) and 53.80/51.31 (volume-rank), while the existence token has a noticeably larger effect. Our practical observation is that many low-existence-score slots after denoising are near-duplicates of high-score primitives. This suggests that the existence score already helps suppress overlapping or redundant primitives to some extent.
>
> **3. On baseline choice and larger-scale evaluation.** We agree that stronger DiT-style denoisers are very promising, especially at larger scale. For this paper, we focus on plain unconditional 3D shape generation, following recent these works with standard setting of ShapeNet, to isolate the role of the representation. Since DoSs works in a compact and explicit token space, it is naturally efficient and controllable. And we believe these advantages will become even clearer in future larger-scale and multimodal conditional settings (such as image-to-3D, text-to-3D), which we highlight as future work.
>
> **4. On robustness of parameters.** To better understand parameter sensitivity, we conducted a simple perturbation study during denoising. For each parameter group (shape, size, rotation, position, existence), we randomly added 10\% noise to that group only, and measured the final deviation from the no-noise output using Manhattan distance.
>
> | Parameter group | Dim | Total Manhattan deviation | Deviation per dimension |
> | --- | --- | --- | --- |
> | Shape | 2 | 153.90 | 76.95 |
> | Size | 3 | 104.20 | 34.73 |
> | Rotation | 6 | 138.78 | 23.13 |
> | Position | 3 | 132.40 | 44.13 |
> | Existence | 1 | 101.59 | 101.59 |
>
> From this preliminary result, the denoising process appears most dependent on coarse structural factors, especially whether a primitive exists and what basic shape it has, while being relatively less sensitive to size and rotation perturbations. We will clarify this as an interesting observation in discussion.
>
> **5. On self-embedded generated superqiadrics.** Since DoSs generates explicit superqiadrics, this can be naturally strengthened with a simple post-processing step such as overlap-based deduplication/NMS over highly overlapping superquadrics, which we expect would improve the cleanliness and editability of the generated structure.
>
> **KeyQuestion: On scaling to larger diffusion states.** We thank the reviewer for this thoughtful question. Intuitively, increasing the number of SQs should improve representational capacity, especially for more complex objects. At the same time, for our current ShapeNet unconditional generation setting, we believe $K_{max}=128$ is a reasonable choice: as shown in following table, the median SQ counts are only 13/19/13, the P90 values are 17/36/38, and even the observed maxima are 52/123/100, all within the current budget.
>
> | Category | Mean SQs / shape | Median SQs / shape | P90 SQs / shape | Max SQs / shape |
> | --- | --- | --- | --- | --- |
> | Airplane | 12.98 | 13 | 17 | 52 |
> | Chair | 21.08 | 19 | 36 | 123 |
> | Car | 18.64 | 13 | 38 | 100 |
>
> *P90 means the 90th-percentile SQ count: 90% of shapes in that category use no more than this many SQs.*
>
> We agree that larger diffusion states will likely become more useful in future extensions to more complex datasets and richer conditional tasks such as text-to-3D or image-to-3D. Beyond simply increasing $K_{max}$ in a fixed-length diffusion model, we are also interested in exploring flexible-length SQ generation, for example token-by-token autoregressive generation. Since limited rebuttal space, we will include a more comprehensive table and discussion of SQ-count statistics and scaling behavior in revision.

---

> > ### Author Rebuttal · Reviewer_mpMf · 2026-04-02
> >
> > After carefully reading the authors' rebuttal, I found that most of the concerns have been addressed. Although the current exploration of the proposed DoSs is limited to traditional benchmarks with a small sample size, I think the idea is novel and has a lot of potential in advancing 3D generation. However, these potentials are difficult to verify in the short period of rebuttal. After consideration, I decided to maintain my positive recommendation and suggest an accept to this paper.

---

> > > ### Author Response · Authors · 2026-04-04
> > >
> > > Thank you very much for the careful follow-up and for reading our rebuttal closely. We greatly appreciate your positive recommendation and your thoughtful recognition of both the novelty of the idea and its potential for advancing efficient and controllable 3D generation. We also appreciate your balanced view regarding the current scope of the experiments, and we will further strengthen the discussion of these limitations and future directions in the revision.

---

### Official Review · Reviewer_BHkw · 2026-03-14

**Soundness:** 2
**Presentation:** 3
**Significance:** 2
**Originality:** 2
**Overall Recommendation:** 4
**Confidence:** 4

**Summary:**

The paper proposes diffusion over superquadric primitives. The key technical contribution is formulation of parameterization of superquadrics in a form that is amenable to diffusion: set of parameters, existence score that defines superquadric existence; diffusion amenable 3d rotation parameterization; canonic ordering of primitives. Experiments are done on chair, airplane and car categories of the ShapeNet dataset.

**Compliance With Llm Reviewing Policy:**

Affirmed.

**Final Justification:**

After reading other reviews and rebuttal, I have adjusted my rating to weak accept given the fact that other reviewers feel positive about the paper. I strongly recommend including additional experiments and discussion in final version of the paper.

**Key Questions For Authors:**

-Have the authors tried to compare the proposed method to superquadric decomposition (see weaknesses) of shapes generated by strong generative 3d model?
-Have authors tried cross-category training and if yes, how well does it perform?

**Limitations:**

Important limitation is not listed: not clear how model performs in cross-category training

**Strengths And Weaknesses:**

**Strengths**
+ Technical insights how to run diffusion on super-quadrics: parameterization for diffusion; existence score; diffusion of 6d rotations;
+ Detailed ablation for each element of the method;
+ Method is fast to train and run prediction on.

**Weaknesses**
- Very weak baselines. The latest baselines are from 2023 and 3D shape generation fields have made significant advances since then (3DILG, 3DShape2VecSet, TRELLIS 1, TRELLIS 2, GEM3D, CLAY, Michelangelo, MeshGPT, Wonder3D and many others).
- Method is not evaluated against a simple baseline: strong generative 3d model (see above) that is decomposed in superquadrics primitives via Paschalidou et. al [1] or similar method.
- Poor qualitative results. Proposed method only generates simple shapes with disconnected and misaligned quadrics;
- Very limited evaluation in terms of shape categories. According to the authors, training takes 2 hours per category which means that training for 10 categories would take less than 24 hours;
- It is unclear how the method performs in cross-category training and evaluation.

[1] Paschalidou D, Ulusoy AO, Geiger A. Superquadrics revisited: Learning 3d shape parsing beyond cuboids. InProceedings of the IEEE/CVF conference on computer vision and pattern recognition 2019 (pp. 10344-10353).

---

> ### Author Rebuttal · Authors · 2026-03-31
>
> We thank the reviewer for the constructive feedback. We are encouraged that the reviewer recognizes the paper’s main technical contributions.
>
> **On cross-category training and generalization**, we thank the reviewer for emphasizing it and we notice that the evidence was not sufficiently visible in the main paper. The caption of Table 1 (line 278) already points out additional multi-class training results in Appendix A. There, we report a single DoSs model jointly trained on Chair, Airplane, and Car via category MLP conditioning. The resulting 1-NNA/COV scores are 55.70/50.19 for Chair, 60.20/45.36 for Airplane, and 61.11/47.78 for Car, and the joint model is trained for 6 hours on one RTX 4090. These results indicate that DoSs is not restricted to strictly category-specific training. In the revision, we will discuss cross-category behavior more explicitly as part of the evaluation.
>
> **On weak baselines**, we agree that the comparison can be better discussed. At the same time, we believe that the baselines used in the paper are appropriate for the problem setting and the paper’s main claim.
>
> First, several methods mentioned in the review target different settings from ours. Our paper studies plain unconditional category-level 3D shape generation, evaluated by 1-NNA/COV of standard ShapeNet setting (Chair, Airplane, and Car) where we follow prior unconditional 3D shape generation works. In contrast, methods such as CLAY, Michelangelo, Wonder3D, and TRELLIS primarily focus on conditional settings such as text-to-3D or image-to-3D, often with different output modalities, training scales, and evaluation protocols. A direct numerical comparison would therefore conflate diffusion state strength with conditioning strength, data scale, and output format.
>
> Second, among the mentioned methods, a subset are truly close baselines for our exact setting. In particular, 3DILG (2022) and 3DShape2VecSet (2023) are not actually newer than the dense-state baselines already included in our comparison such as TIGER (2024). We do agree, however, that GEM3D and MeshGPT, are relevant references and deserve more explicit discussion. For transparency, we summarize below the reported inference times and the available 1-NNA/COV results under the cited settings.
>
> | Method | Inference (s/shape) | Chair 1-NNA | Chair COV | Airplane 1-NNA | Airplane COV | Car 1-NNA | Car COV |
> | --- | --- | --- | --- | --- | --- | --- | --- |
> | 3DILG | - | - | 61.4 | - | 56.2 | - | 41.9 |
> | 3DShape2VecSet | - | - | 51.9 | - | 46.5 | - | 44.0 |
> | GEM3D | ~30 | - | 61.3 | - | 64.3 | - | 42.8 |
> | MeshGPT | 30-90 | 75.51 | 43.28 | - | - | - | - |
> | DoSs (ours) | 0.6 | 53.80 | 51.31 | 61.92 | 46.77 | 57.50 | 49.26 |
>
> As the table shows, DoSs is more efficient than both GEM3D and MeshGPT, while the quality comparison is only partial because GEM3D does not report 1-NNA and MeshGPT reports only Chair. These methods are therefore relevant references, but not uniformly stronger, fully matched baselines for our protocol.
>
> **On poor qualitative results,** we agree that the current samples are less visually smooth at local primitive boundaries than dense-state generators. However, this reflects the core trade-off of DoSs rather than a contradiction of the paper’s claim: DoSs prioritizes a compact explicit representation that offers substantially higher efficiency and controllability than dense generators, even if this comes at some cost in pure visual smoothness. Accordingly, Table 1 compares methods not only in quality, but also in diffusion-state size and inference time across different denoising spaces. Under this perspective, DoSs denoises only ~7 KB and runs at 0.6 s/shape with controllability, while remaining competitive on standard 1-NNA/COV metrics after point-cloud decoding. We will revise the wording to make this scope clearer and to better align the qualitative discussion with the actual contribution of the paper.
>
> We also thank the reviewer for suggesting a “strong generator + superquadric decomposition” baseline. We agree that this would be a useful additional reference point. However, such a baseline measures a two-stage system, and its final performance depends on both the generator and decomposition efficiency-quality. Our method instead learns the generative process directly in primitive which highlights a different diffusion operating point. We also fully agree that the current DDPM + U-Net is not the strongest possible generator. The contribution of this paper is therefore not a claim of the strongest methods, but a new formulation for diffusion-based 3D shape generation: performing diffusion over a compact, explicit primitive space instead of dense geometry, with the resulting gains in natural controllability and efficiency. We will clarify this more explicitly in the revision, and also highlight that stronger performance is likely to come from better decomposition/fitting, stronger generators, and hybrid pipelines in the future work.

---

> > ### Author Rebuttal · Reviewer_BHkw · 2026-04-03
> >
> > I appreciate the work authors’ did for the rebuttal.
> >
> > Majority of my concerns weee addressed but I am still concerned that proposed approach might not perform as well as superquadrics  decomposition of good generative model.
> >
> > Given the fact that other reviewers are excited about this work I can adjust my score to weak accept.

---

> > > ### Author Response · Authors · 2026-04-04
> > >
> > > Thank you for the follow-up and reconsidering the score as weak accept.
> > >
> > > We are glad that most concerns were addressed. Regarding the suggested two-stage system (strong dense generator followed by superquadric decomposition), we agree that it is a meaningful additional reference point. At the same time, based on our observations, DoSs benefits from direct primitive-space controllability and sub-second inference, whereas a two-stage pipeline would typically involve minute-level processing and only indirect control through post-hoc decomposition (for example, such two-stage system is not able to support constrained denoising for geometric-aware design). In this sense, the two approaches target different operating points.
> > >
> > > For quality, we would also like to note that 1-NNA and COV primarily assess the distributional match and diversity of the generated set, rather than pure per-shape reconstruction fidelity. Therefore, even if a strong dense generator produces visually strong samples(for example, the reported DiT-3D), a subsequent decomposition stage may still degrade these metrics: the additional fitting/decomposition error, together with sampling noise, can push shapes away from the original generator’s distribution. For this reason, we believe a two-stage system is unlikely to preserve the full 1-NNA/COV performance of the one-stage dense generators(our baselines). More broadly, our claim is not the strongest possible generator, but that DoSs introduces a direct primitive-space diffusion formulation with favorable controllability–efficiency–quality trade-offs. We will clarify this scope more explicitly in the revision.

---

### Decision · Program_Chairs · 2026-04-30

**Decision:**

Accept (regular)

**Comment:**

This paper got three weak accept and one accept. All reviewers acknowledged the technical contributions of the paper. Authors provided a detailed rebuttal that well addressed the concerns of the reviewers.